# Inter-dependent apical microtubule and actin dynamics orchestrate centrosome retention and neuronal delamination

Ioannis Kasioulis, Raman M Das[†]*, Kate G Storey*

Division of Cell and Developmental Biology, School of Life Sciences, University of Dundee, Dundee, United Kingdom

*For correspondence:
raman.das@manchester.ac.uk
(RMD);
k.g.storey@dundee.ac.uk (KGS)

Present address: [†]Division of
Developmental Biology and
Medicine, School of Medical
Sciences, Faculty of Biology
Medicine and Health,
Manchester Academic Health
Science Centre, University of
Manchester, Manchester, United
Kingdom

Competing interests: The
authors declare that no
competing interests exist.

Reviewing editor: David J
Solecki, St Jude Children's
Research Hospital, United States

**Abstract** Detachment of newborn neurons from the neuroepithelium is required for correct neuronal architecture and functional circuitry. This process, also known as delamination, involves adherens-junction disassembly and acto-myosin-mediated abscission, during which the centrosome is retained while apical/ciliary membranes are shed. Cell-biological mechanisms mediating delamination are, however, poorly understood. Using live-tissue and super-resolution imaging, we uncover a centrosome-nucleated wheel-like microtubule configuration, aligned with the apical actin cable and adherens-junctions within chick and mouse neuroepithelial cells. These microtubules maintain adherens-junctions while actin maintains microtubules, adherens-junctions and apical end-foot dimensions. During neuronal delamination, acto-myosin constriction generates a tunnel-like actin-microtubule configuration through which the centrosome translocates. This movement requires inter-dependent actin and microtubule activity, and we identify drebrin as a potential coordinator of these cytoskeletal dynamics. Furthermore, centrosome compromise revealed that this organelle is required for delamination. These findings identify new cytoskeletal configurations and regulatory relationships that orchestrate neuronal delamination and may inform mechanisms underlying pathological epithelial cell detachment.

DOI: https://doi.org/10.7554/eLife.26215.001

## Introduction

Delamination involves extraction of a cell from within a proliferative tissue. It is a fundamental process underlying epithelial tissue morphogenesis that is linked to cell state change during normal differentiation and also to cancer cell dispersal. Cells undergoing neuronal differentiation delaminate from the proliferative domain of the neuroepithelium and this involves loss of adhesion between neighbouring cells at the ventricular surface. This process is required for correct neuron placement (*Kriegstein and Noctor, 2004*; *Singh and Solecki, 2015*), and this in turn is necessary for subsequent formation of functional neuronal circuitry. Neuronal delamination defects are collectively known as periventricular heterotopias and lead to a spectrum of deficits including epilepsy, dyslexia and intellectual disability (*Lian and Sheen, 2015*; *Passarelli and Moreira, 2014*).

A genetic basis for human periventricular heterotopia has been mapped to the actin cross-linking protein, FilaminA and the ADP-ribosylation factor guanine exchange factor 2 ARFGEF2/BIG2 (*Lian and Sheen, 2015*). The interaction between these proteins has implicated them in vesicle trafficking and stability/turnover of cell adhesion proteins (*Zhang et al., 2013*; *Zhang et al., 2012*). These data are consistent with work linking mutation of cadherins FAT4 and DCHS1 with a periventricular heterotopia phenotype (*Badouel et al., 2015*; *Cappello et al., 2013*). Experiments in animal models implicate further regulators of cell adhesion in neuronal delamination, including Slit/Robo, which also acts by attenuating N-cadherin activity (*Wilsch-Bräuninger et al., 2016*; *Wong et al., 2012*) (*Borrell et al., 2012*). Overall, many such proteins associated with apically localised adherens

**eLife digest** The brain and spinal cord begin as a tube that runs the length of the developing embryo. This tube made from cells called neural progenitors, which can divide to generate adult nerve cells. As nerve cells are born they detach from their neighbours, in a process called delamination before migrating away.

Though the delamination of nerve cells is important for the formation of the nervous system, scientists do not fully understand how proteins inside cells work together to release the newborn nerve cell from its neighbours. Two major components of the process are proteins called actin and tubulin, which form complex structures known as acto-myosin cables and microtubules respectively. Acto-myosin cables must contract during delamination, but the role of the microtubules is unclear.

Kasioulis et al. examined the microtubules in chick and mouse neural tube cells during delamination using fluorescent labels to mark key molecules and small molecule inhibitors to selectively block different activities. A combination of live tissue and super-resolution imaging were used to reveal the dynamics of the delamination process.

The experiments revealed a wheel-like configuration of microtubules that lined up with the acto-myosin cable. Actin maintained the microtubules, which in turn maintained the acto-myosin cable. As newborn neurons delaminated, the actin cable constricted and the microtubules condensed, forming a tunnel that allowed a structure that organises the microtubules – the centrosome – to move, and the cell to detach. A protein called Drebrin, which links actin to microtubules, was identified as a potential coordinator of the process.

These findings not only further our understanding of nervous system development, but may also shed light on the development of human diseases. Failure of delamination can lead to a spectrum of disorders, including epilepsy, dyslexia and intellectual disability. Cell detachment is also important in other developmental processes, as well as in the spread of cancer cells.

DOI: https://doi.org/10.7554/eLife.26215.002

junctions (AJs) have been linked to the delamination process (*Cappello et al., 2006*; *Imai et al., 2006*; *Kadowaki et al., 2007*; *Singh and Solecki, 2015*; *Stocker and Chenn, 2009*; *Stocker and Chenn, 2015*). AJs are required for the integrity of the entire neuroepithelium and so delamination defects and precocious neuronal differentiation are most readily seen following cell-autonomous deletion of associated proteins (*Stocker and Chenn, 2009*; *Stocker and Chenn, 2015*; *Woodhead et al., 2006*; *Zhang et al., 2010*). However, despite such manipulations we know little about the cell biological mechanisms that mediate delamination as AJs disassemble.

Recent high-resolution live tissue-imaging of chick spinal cord has revealed that detachment of the newborn neuron from the ventricle is mediated by a novel cell sub-division mechanism, apical abscission, which leads to shedding of the apical tip of the cell (*Das and Storey, 2014*; *Das and Storey, 2014b*). The apical poles of neuroepithelial cells are characterised by the presence of a contractile sub-apical acto-myosin cable which is mechanically and biochemically linked to cadherin-containing AJs (*Abe and Takeichi, 2008*; *Marthiens and ffrench-Constant, 2009*; *Maul et al., 2003*; *Miyamoto et al., 2015*). Apical abscission is triggered by acto-myosin cable constriction following attenuation of N-cadherin; this process is blocked by N-cadherin mis-expression (*Das and Storey, 2014*) while repression of *N-cadherin* transcription downstream of the neurogenic transcription factor cascade, which promotes neuronal differentiation, leads to loss of cell–cell contact at the ventricular surface (*Rousso et al., 2012*). Similar transcription factor activity that promotes neuronal delamination in the brain involves regulation of cadherin/apical polarity proteins by Snail superfamily members (and others) (*Acloque et al., 2009*; *Itoh et al., 2013*; *Singh et al., 2016*; *Singh and Solecki, 2015*). Importantly, such proteins also induce cell-cell detachment during epithelial to mesenchymal transition in other tissues and in oncogenic contexts suggesting operation of shared downstream cell biological mechanisms.

In some respects, apical abscission resembles cytokinesis, where a contractile acto-myosin ring generates the forces that separate the two daughter cells. A key structure regulating this cytokinetic ring is the central spindle, which consists of an array of antiparallel microtubules as well as de novo synthesized microtubules (*Fededa and Gerlich, 2012*). This raises the possibility that microtubules

regulate the apical acto-myosin cable in neuroepithelial cells during delamination. Like actin, microtubules are also associated with AJs (*Bellett et al., 2009*; *Ligon et al., 2001*; *Meng et al., 2008*; *Stehbens et al., 2006*) and cadherin-mediated adhesion can recruit and stabilize microtubules (*Stehbens et al., 2006*; *Waterman-Storer et al., 2000*). Conversely, AJs are destabilized by microtubule de-polymerisation in a variety of cell types in vitro (*Mary et al., 2002*; *Yap et al., 1995*). This microtubule support for AJs involves kinesin-based transport of cadherin containing vesicles (*Mary et al., 2002*) and specifically in neuroepithelial cells by the KIF3 motor complex (*Teng et al., 2005*), although this transport role is context dependent (*Stehbens et al., 2006*). Furthermore, microtubule de-polymerisation or stabilisation can block AJ disassembly (*Ivanov et al., 2006*) suggesting a more complex relationship between cadherin supply and AJ integrity. Little is known about the organisation of microtubules and their relationship with actin and AJs in the neuroepithelial cells or how they might regulate neuronal delamination.

A relationship between regulation of AJs and cell cycle exit is suggested by findings that link AJs to mitogenic signalling via Notch and Wnt pathways (*Hatakeyama et al., 2014*; *Zhang et al., 2010*). In the chick spinal cord, apical abscission is preceded by dis-assembly of the primary cilium (*Das and Storey, 2014*) and loss and or retraction of ciliary membrane is also associated with delaminating zebrafish retinal neuroblasts (*Lepanto et al., 2016*). Mediators of the mitogenic Sonic hedgehog pathway are processed into activated forms in the primary cilium (*Guemez-Gamboa et al., 2014*; *Kim et al., 2009*) and so this may be a further way in which cell biological mechanisms associated with delamination link this process to cell state change. Following cilium disassembly, the centrosome is retained in the withdrawing neuronal cell process while ciliary and apical membrane are shed (*Das and Storey, 2014*). Centrosome retention is then critical for subsequent neuronal differentiation: for neuronal migration to form the cortical plate (*Higginbotham and Gleeson, 2007*; *Tsai and Gleeson, 2005*; *Xie et al., 2003*), as a microtubule organising centre during axonogenesis (*de Anda et al., 2005*; *Zmuda and Rivas, 1998*), and in defining where dendrites will elongate (*Puram and Bonni, 2013*; *Puram et al., 2011*), although this is context dependent (*Kuijpers and Hoogenraad, 2011*). The role of the centrosome in delamination and the mechanism that ensures its retention in newborn neurons are, however, not known. Here, we use live-tissue imaging and super-resolution microscopy to elucidate the cytoskeletal architecture of the apical end-foot of neuroepithelial cells and to dissect the regulatory relationships which underpin cytoskeletal dynamics underlying neuronal delamination.

## Results

### A wheel-like microtubule configuration in the neuroepithelial cell apical end-foot

To localise microtubules within neuroepithelial cells, we carried out immunocytochemistry in sections of chick spinal cord (at Hamburger and Hamilton stage HH17-8) (*Hamburger and Hamilton, 1951*) to detect the stable microtubule marker, acetylated α-tubulin (*Perdiz et al., 2011*), the AJ-associated protein N-cadherin and the actin cytoskeleton using phalloidin. The microtubule cytoskeleton was enriched apically and overlapped with the actin cable and the AJs (*Figure 1A–A''''*). Closer examination of microtubule architecture in neuroepithelial cell apical end-feet using en face imaging, revealed a sub-apical ring-like structure ($2.57 \pm 0.5$ μm in diameter, 21 cells, in 2 explants from 2 embryos) and associated microtubules radiating from the centrosome of the primary cilium, identified by γ-tubulin and IFT88, respectively (*Figure 1B–B''*, *C–C'* and *D–D'*). A similar microtubule configuration was observed by en face imaging of the ventricular surface in E12.5 mouse spinal cord and cortex (*Figure 1E*, in 4 explants from 2 embryos) indicating conservation of this apical microtubule architecture across species and different regions of the central nervous system.

To place these apical microtubules in the context of known apical sub-cellular organisation, we next captured the three-dimensional relationship between the alpha-tubulin-labelled microtubules, the actin cable and associated N-cadherin-containing AJs, imaging from the apical surface of the chick spinal cord in en face orientation (*Figure 1F*, *Figure 1—video 1*, n = 167 cells, in 4 explants from 4 embryos). The alignment of actin and tubulin was then measured at the Z-level defined by N-Cadherin localisation; this revealed actin-tubulin co-alignment in the majority of cells (71%) (*Figure 1—figure supplement 1*, 31 cells in 3 explants from 3 embryos). A subset of microtubules was

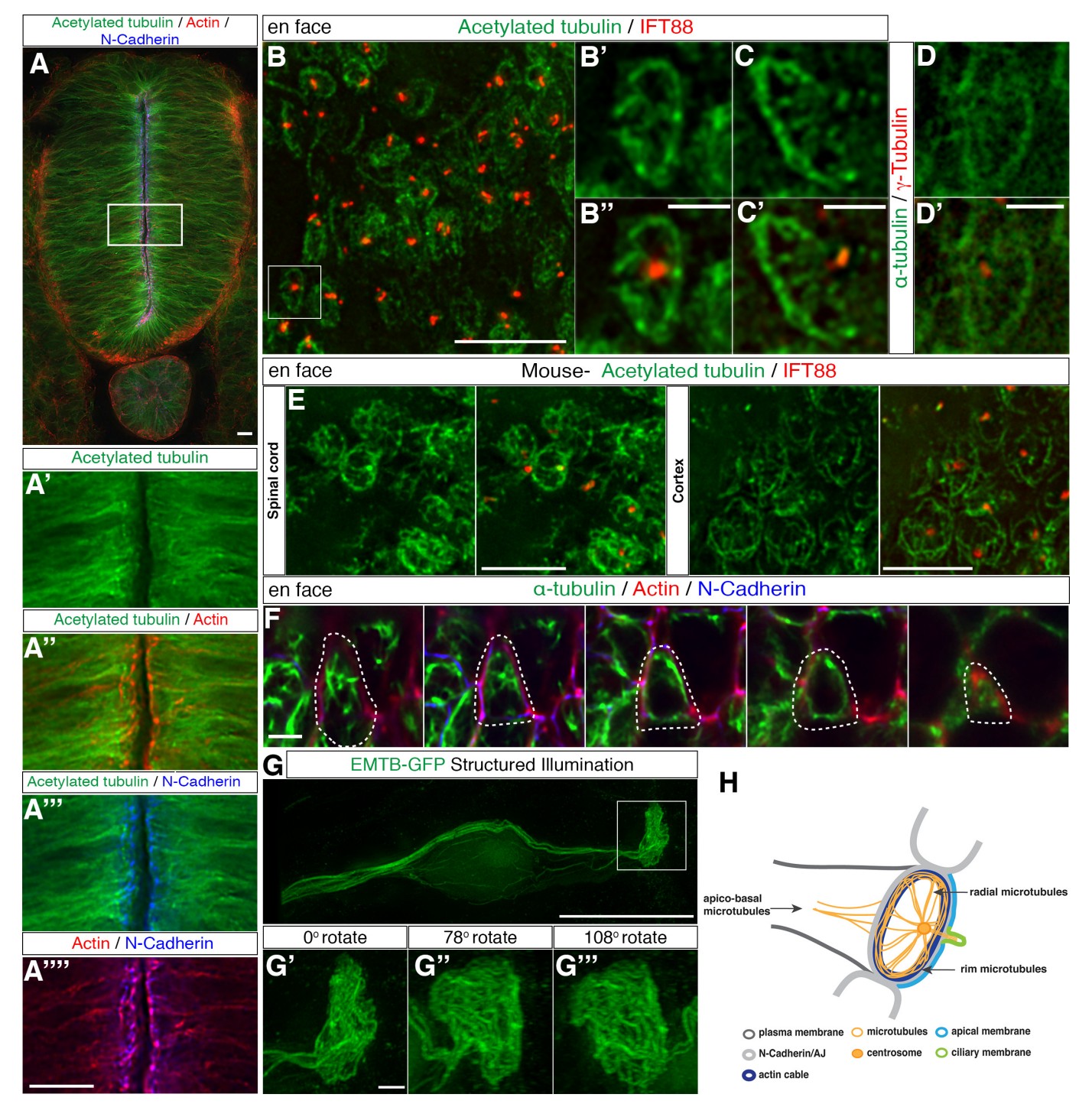

**Figure 1.** Characterisation of the sub-apical microtubule architecture. (**A**) Representative image of a 3-day-old chick embryo neural tube stained with acetylated α-tubulin, phalloidin and N-Cadherin. (**A'–A''''**) Magnification of the boxed region in (**A**). (**B**) En face imaging of neuroepithelial end-feet with acetylated α-tubulin and IFT88. (**B'–B''**) Magnification of boxed region in (**B**). (**C–C'**) Another example as in (**B'**). (**D–D'**) End-foot stained with α-tubulin and γ-tubulin. (**E**) En face imaging of E12.5 mouse embryo spinal cord and cortex stained with acetylated α-tubulin and IFT88. (**F**) Stills of a neuroepithelial cell (dotted lines show cell outline) en face imaging from apical to more basal (left to right). Tissue explant stained for α-tubulin, N-Cadherin and phalloidin. (**G**) Neural progenitor cell expressing EMTB-GFP (and nuclear localised GFP from pCIG-Neurog2) imaged with SIM. The boxed region was magnified in (**G'–G'''**). Three different angles off the boxed region in G generated by 3D reconstruction. (**H**) Diagram of microtubule

*Figure 1 continued on next page*

*Figure 1 continued*

organization at the apical end-feet and relationship with the acto-myosin ring and the AJs. For all figures, images were captured by wide-field microscopy, unless otherwise stated. Scale bars, (A) (B) (E) (G) (A'–A'''') 10 µm, (B'–B'') (C–C') (D–D') (F) (G'–G''') 2 µm.

DOI: https://doi.org/10.7554/eLife.26215.003

The following video, source data, and figure supplement are available for figure 1:

**Source data 1.** Actin-tubulin co-alignment at the apical adhesion belt level.

DOI: https://doi.org/10.7554/eLife.26215.005

**Figure supplement 1.** Actin-tubulin co-alignment at the adhesion belt level.

DOI: https://doi.org/10.7554/eLife.26215.004

**Figure 1—video 1.** Apico-basal Z-stack series across the apical microtubules and sub-apical actin cable and N-Cadherin based adherens junctions; this video is related to *Figure 1F*.

DOI: https://doi.org/10.7554/eLife.26215.006

**Figure 1—video 2.** 3D structured illumination reconstruction of EMTB-GFP mis-expression at the neuroepithelial cell apical end-foot; this video is related to *Figure 1G–G'''*.

DOI: https://doi.org/10.7554/eLife.26215.007

also observed to extend basal to the actin/N-cadherin junctional region deep into the cell-process (*Figure 1F*).

To capture the overall microtubule configuration in individual cells, we next mis-expressed a GFP-tagged microtubule binding protein MAP7/Ensconsin (EMTB-GFP) (*Bulinski et al., 1999*) along with a plasmid expressing the proneural factor *Neurog2* (pCAGGS-Neurog2_IRES-nucGFP, pCIG-Neurog2) to promote neuronal differentiation (*Ma et al., 1996*) in a scattering of cells in the developing chick spinal cord (HH 10–12). Mis-expression of EMTB at high levels can stabilise microtubules (*Bulinski et al., 1999*) and this facilitated use of structured illumination microscopy (SIM) to generate super-resolution images of extensive microtubule structures within neuroepithelial cells. Analysis of such individual cells in transverse embryo slices revealed a more elaborate microtubule meshwork and also continuity between sub-apical microtubules and apico-basal orientated microtubules that extend towards and around the cell nucleus (*Figure 1G–G'''*, and *Figure 1—video 2*, 4 cells from 2 embryos). Together these two and three-dimensional analyses suggest the presence of a sub-apical wheel-like microtubule organisation, composed of radial microtubules emanating from the centrosome and rim microtubules aligned with actin/N-cadherin, which is further continuous with apico-basal microtubules that extend the length of the cell, summarised in *Figure 1H*.

## The centrosome nucleates microtubules which radiate towards and extend along the actin cable

To substantiate the centrosomal origin of the radial and rim microtubules, we next used live tissue imaging to monitor microtubule nucleation patterns in the apical end-foot. This involved mis-expression of PACT-TagRFP to label the centrosome and EB3-GFP to identify microtubule plus-ends (*Gillingham and Munro, 2000*; *Stepanova et al., 2003*) in chick spinal cord and monitoring cell behaviour in an adapted en face version of ex-vivo embryo slice cultures using high-resolution wide-field microscopy (*Das et al., 2012*). Tracking the trajectory of EB3-GFP comets revealed that radial microtubules emanate in an evenly spaced fashion from the centrosome of the primary cilium in the end-foot (*Figure 2A*; *Figure 2—video 1*, 51 cells in 3 explants from 3 embryos). By combining EB3-GFP and F-tractin-mKate2 to monitor the relationship between these microtubules and the actin cable, we further observed some EB3-GFP comets running along the actin cable (*Figure 2B*; *Figure 2—video 2*, 95 cells in 4 explants from 4 embryos). To quantify this relationship, we followed the 2D trajectories of EB3-GFP comets and measured the EB3-GFP/F-tractin-mKate2 inter-peak distance over time. This analysis indicated a close alignment of polymerising microtubules with the actin belt (*Figure 2D–F*; *Figure 2—video 3*; trail tracking: 10 cells in 3 explants from 3 embryos). Tracking comet movements also delineated microtubule shapes and revealed that radial microtubules bend as they reach the periphery and turn to run along the actin cable (*Figure 2C*; *Figure 2—video 4*, 12 cells in 4 explants from 4 embryos). These dynamic data further support the case for a wheel-like organisation of apical microtubules, demonstrate that the centrosome is the source of both radial and rim microtubules and confirm the close alignment of rim microtubules with the actin cable and AJs.

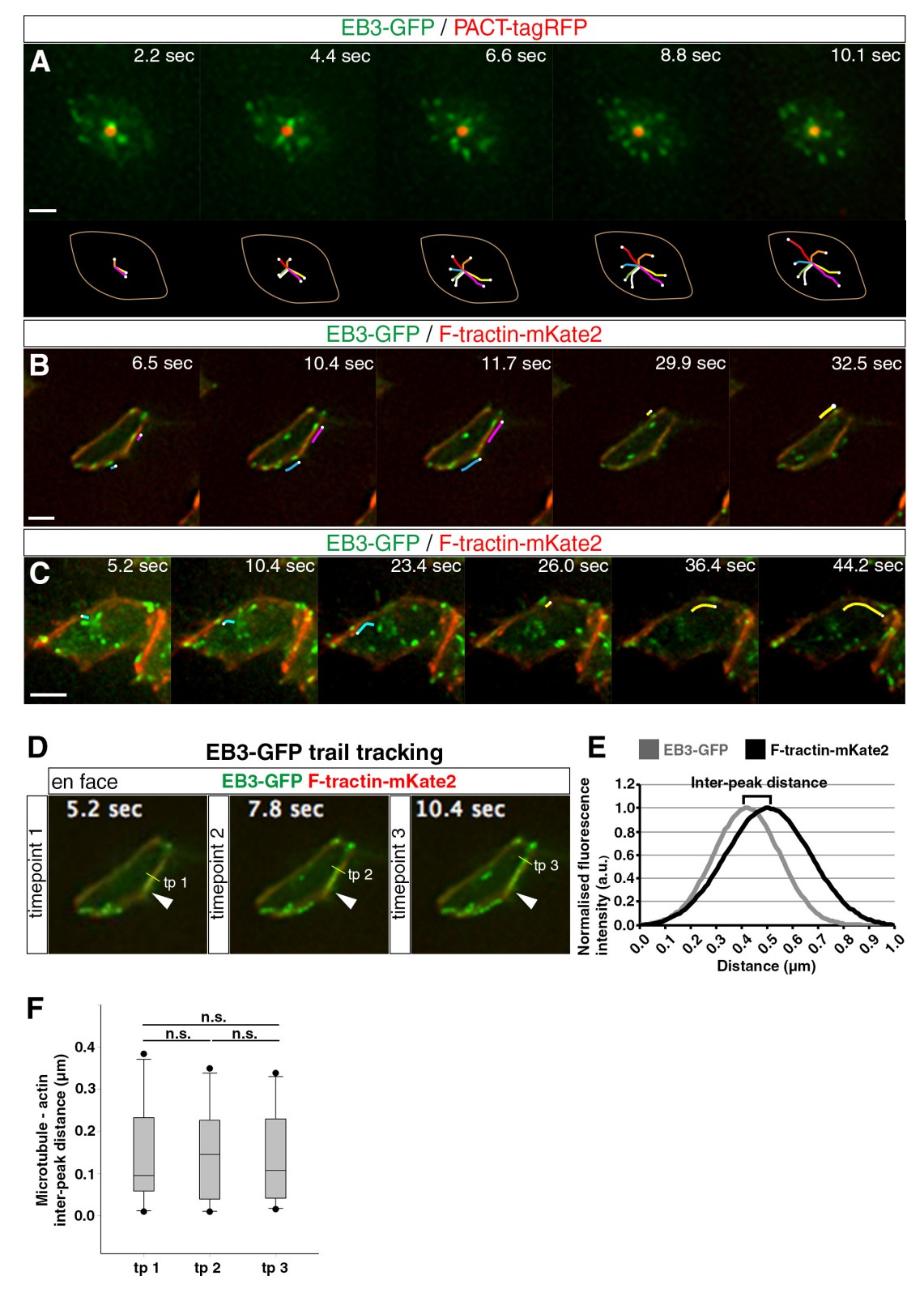

**Figure 2.** Microtubule dynamics at the apical end-foot and alignment with the actin belt. (A) Microtubule nucleation from the centrosome. The apical end-foot outline and tracking of EB3-GFP comets over time are shown below. (B) Movement of polymerising microtubules along the actin cable. Lines track movement of two EB3-GFP comets. (C) Microtubules nucleated from the centrosome bend and travel along the actin cable. Lines follow the movement of two EB3-GFP comets. (D) Trail tracking of EB3-GFP comets over time along the F-tractin-mKate2 belt. Three timepoints are shown. The

*Figure 2 continued on next page*

*Figure 2 continued*
arrowhead represents the starting point of EB3-GFP comet movement. The yellow line shows its position at different timepoints and the method for the measurement of fluorescence intensity at that particular point for both channels. (E) Example of fitted Guassian curves for the calculation of inter-peak distance between the two channels. For the purpose of this example, both fitted fluorescence intensity calculations were normalised from 0 to 1. (F) Box-plots of the microtubule (EB3-GFP)- actin (F-tractin-mKate2) inter-peak distance over time (paired t-test: tp 1 vs tp 2, p=0.84; tp 2 vs tp 3, p=0.72; tp 1 vs tp 3, p=0.96). Scale bars, (A) (B) (C) 2 μm.
DOI: https://doi.org/10.7554/eLife.26215.008
The following video and source data are available for figure 2:

**Source data 1.** EB3-GFP_F-tractin-mKate2 inter-peak distance.
DOI: https://doi.org/10.7554/eLife.26215.009
**Figure 2—video 1.** Microtubule nucleation from the apical centrosome; this video is related to *Figure 2A*.
DOI: https://doi.org/10.7554/eLife.26215.010
**Figure 2—video 2.** Microtubule movement along the actin cable; this video is related to *Figure 2B*.
DOI: https://doi.org/10.7554/eLife.26215.011
**Figure 2—video 3.** Trail tracking of EB3-GFP comets; this video is related to *Figure 2—video 2* and *Figure 2D*.
DOI: https://doi.org/10.7554/eLife.26215.012
**Figure 2—video 4.** Microtubule bending at the actin cable; this video is related to *Figure 2C*.
DOI: https://doi.org/10.7554/eLife.26215.013

## Microtubules maintain adherens junctions, while actin maintains microtubules, adherens junctions and apical end-foot dimensions

To test the regulatory relationships between apical microtubules, actin and AJs, we next assessed the consequences of microtubule depolymerisation following exposure to Nocodazole for 1 hr. This treatment depleted apical microtubules as expected (*Figure 3A, B*) and reduced N-Cadherin at AJs (*Figure 3A', B'*) quantified by fluorescence intensity measurements (*Figure 3C, C'*). Depletion of microtubules also increased distribution of actin within the cell (*Figure 3A", B", D*), however, this did not significantly alter actin levels at the adhesion belt (*Figure 3D, D'*) nor reduce apical end-foot area (*Figure 3A''', B''', E*). These findings indicate that apical microtubules maintain AJs as defined by N-Cadherin levels and that they influence actin localisation, although this did not impact the actin cable nor apical end-foot size.

We next tested the effects of actin depletion on AJs and apical microtubules. Brief exposure (15 mins) to Latrunculin-A which binds actin monomers and so prevents their polymerisation (*Coué et al., 1987*) dramatically reduced apical actin as expected (*Figure 3F, G, H, H'*). This treatment depleted apical microtubules (*Figure 3F", G", J*) and consistent with this also reduced N-Cadherin at AJs (*Figure 3F', G'*) and quantified in *Figure 3I, I'*. Actin depletion additionally led to a decrease in apical end-foot size (*Figure 3F''', G''', K*). These findings indicate that an intact actin cable is required for maintenance of apical microtubule structures as well as AJs in neuroepithelial cells and that the actin cytoskeleton serves to define apical end-foot dimensions.

Together, the above findings uncover a wheel-like organisation of sub-apical microtubules that is nucleated by the centrosome of the primary cilium and which aligns with the actin cable, maintains AJs and stabilises the apical cytoskeleton in neuroepithelial cells of the developing embryo. The tissue analysed at these early stages comprises largely neural progenitors in interphase and so we next addressed how this cytoskeletal configuration alters during neuronal delamination.

## Apical cytoskeletal dynamics in delaminating cells

To assess the apical cytoskeletal configuration in delaminating cells, we next monitored EB3-GFP and F-tractin-mKate2 in cells with a small apical end-foot diameter (typically 1–2.5 μm), characteristic of delaminating cells (*Figure 4A*, *Figure 4—video 1*, 6 cells in 4 explants from 4 embryos). This revealed that microtubule growing tips still emanated towards and along the now constricted actin cable in such cells. This suggests that despite declining N-Cadherin in delaminating cells microtubules remain closely associated with peripherally located actin. To elucidate further the spatial organisation of the cytoskeleton in delaminating cells, we used Stimulated Emission Depletion (STED) microscopy to generate super-resolution images of cells expressing F-tractin-mKate2 and EMTB-GFP (along with the neuronal differentiation gene Neurog2, as above). This revealed a close sub-apical alignment of actin and microtubules in such cells and, observed in 3-dimensions, these two

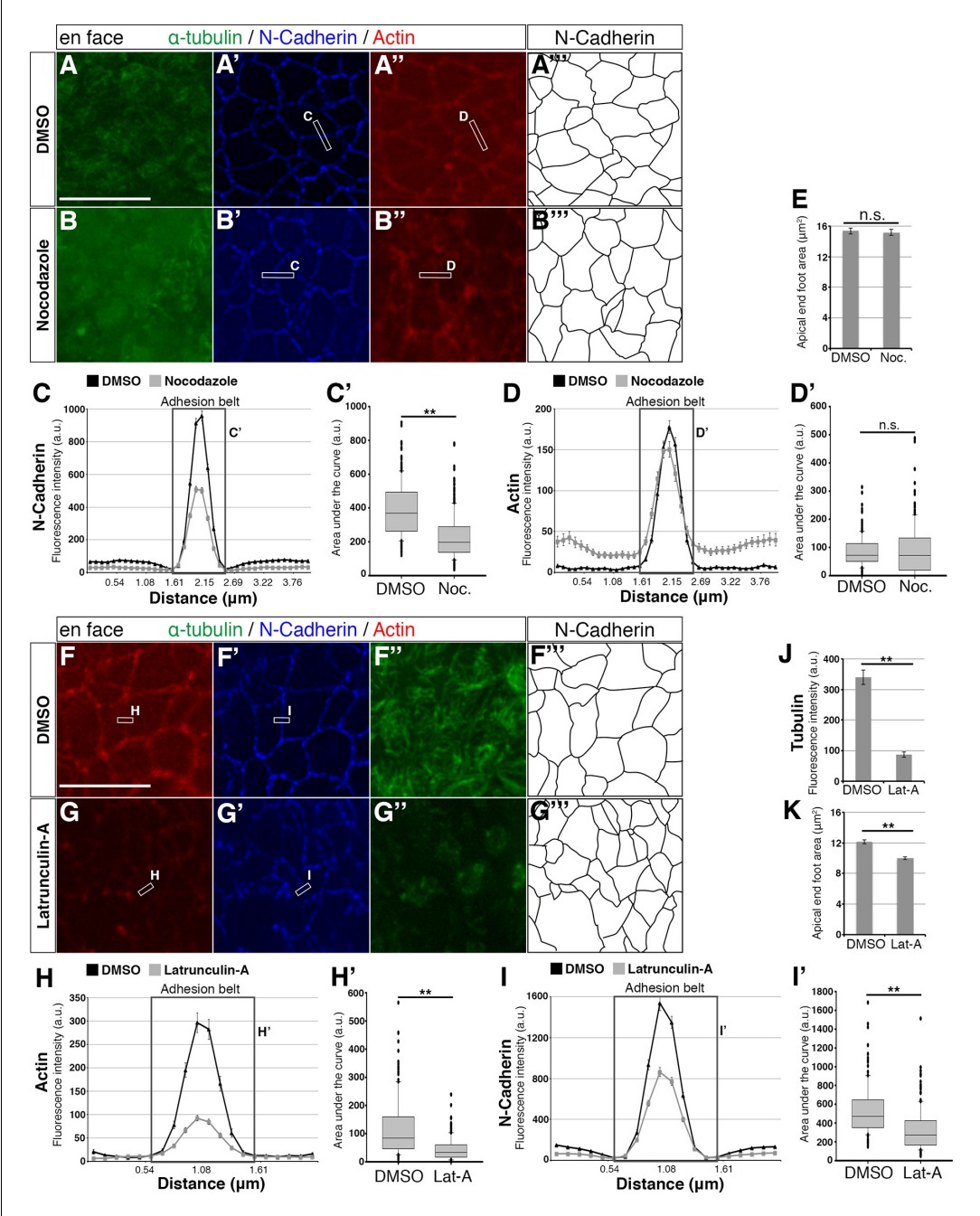

**Figure 3.** Effects of small molecule treatments on the adhesion belt and microtubules. (A – A'', B – B'', F – F'', G – G'') En face imaging of apical end-feet following treatment of chick embryo neural tube explants with Nocodazole (Noc) or Latruncuin-A (Lat-A). Boxed areas indicate how a line is drawn across the adhesion belt for measurement of fluorescence intensity. Letters next to the boxes refer to the corresponding line graphs. (C, D, H, I) Line graphs of normalised fluorescence intensity across the adhesion belt. For Nocodazole a distance of 4 µm and for Latrunculin-A 2 µm was measured. Boxed area represents the adhesion belt and the letter refers to the box plot quantifications from that area. Error bars = SEM. (C', D', H', I') Box plots of the area under the curve (adhesion belt) from the line graphs. The median value, as well as the upper and lower quartiles are represented. T-test, (C') p<0.0001 (DMSO [Nocodazole control]: 210 measurements, 6 explants in 3 experiments; Nocodazole: 270 measurements, 8 explants in 3 experiments), (D') p=0.51 (DMSO [Nocodazole control]: 180 measurements, 6 explants in 3 experiments; Nocodazole: 244 measurements, 8 explants in 3 experiments), (H') p<0.0001 (DMSO [Latrunculin-A control]: 140 measurements, 5 explants in 2 experiments; Latrunculin-A: 213 measurements, 7 explants in 3 experiments) and (I') p<0.0001 (DMSO [Latrunculin-A control]: 140 measurements, 5 explants in 2 experiments; Latrunculin-A: 213 measurements, 7 explants in 3 experiments). When the entire curve is considered in (D), the area of the Nocodazole treatment is statistically larger than that of the DMSO treatment, p<0.0001. (E, K) End-foot area measurements for DMSO and small molecule treatments, as outlined by the N-Cadherin staining (A''', B''', F''', G'''). T-test, (E) p=0.73 (DMSO [Nocodazole control]: 276 measurements in 3 experiments; Nocodazole: 304 measurements in 3

*Figure 3 continued on next page*

*Figure 3 continued*

experiments) and (K) p<0.0001 (DMSO [Latrunculin-A control]: 222 measurements in two experiments; Latrunculin-A: 334 measurements in 3 experiments). Error bars = SEM. (J) Normalised tubulin fluorescence following DMSO or Latrunculin-A treatment. T-test, p<0.0001 (DMSO: 110 measurements in 2 experiments; Latrunculin-A: 205 measurements in 3 experiments). Error bars = SEM, scale bars, 10 μm.

DOI: https://doi.org/10.7554/eLife.26215.014

The following source data is available for figure 3:

**Source data 1.** Nocodazole vs DMSO control.

DOI: https://doi.org/10.7554/eLife.26215.015

**Source data 2.** Latrunculin-A vs DMSO control.

DOI: https://doi.org/10.7554/eLife.26215.016

cytoskeletal components appeared to form a composite tunnel-like configuration (*Figure 4B*, *Figure 4—video 2*, 4 cells from 2 embryos).

To monitor overall microtubule dynamics during delamination in live tissue, spinal cord cells were next co-transfected with EMTB-GFP, pCIG-Neurog2 and mKate2-GPI to label cell membranes and to monitor the changing morphology of individual cells. We then observed neurogenesis in ex-vivo embryo transverse slice cultures as described in *Das et al., 2012*. We monitored cells with moderate levels of EMTB-GFP transfection and observed that the prominent sub-apical EMTB-GFP labelling was highly dynamic and its intensity progressively increased as delamination proceeded. Following completion of abscission, the condensed band of EMTB-GFP was then rapidly lost from the tip of the withdrawing cell-process (*Figure 4C*, *Figure 4—video 3*, 22 cells in 15 slices; in each experiment slices are taken from 2 or 3 embryos, this applies here and in all similar experiments below). This dynamic pattern of enrichment and subsequent loss following abscission is very similar to that we observed previously for actin during this process (*Das and Storey, 2014*). These findings further support the coordinated condensation of apical actin and microtubules during delamination and raised the possibility that apical microtubule re-organisation plays a role in this process.

## Microtubules are required for neuronal delamination

To test whether microtubules are required for neuronal delamination neural tubes were first co-transfected with GFP-GPI and pCIG-Neurog2; following 18 hr of incubation, many transfected cells were found to have adopted a configuration with a basally located nucleus and long cell-process contacting the ventricular surface, indicative of imminent neuronal differentiation. In control DMSO treated slices 19/61 cells (31% in 29 slices) then abscised within 4 hr (*Figure 5A*; *Figure 5—video 1*). However, fewer labelled cells exposed to nocodazole delaminated during this period (8/51 cells, 16% in 35 slices) (*Figure 5B*, *Figure 5—video 2*)(an effective nocodazole concentration (8.5 μM) for this embryo slice culture assay was determined by monitoring mitotic arrest see *Figure 5—figure supplement 1*, *Figure 5—videos 3* and *4*). These data suggest that microtubules are required for delamination. We next used the microtubule stabilising agent taxol (*Jordan and Wilson, 1998*), which reduces microtubule plus end growth (*Kleele et al., 2014*; *Marx et al., 2013*), to determine whether this process relies on dynamic microtubules. The effectiveness of taxol concentration in this embryo slice assay (10 μM) was also first determined using live imaging to assess induction of mitotic arrest (*Figure 5—figure supplement 2*, *Figure 5—video 5*). Cells were transfected as above and cell behaviour monitored following exposure to control DMSO or taxol. While many cells abscised in DMSO treated slices within 6 hr (24/55 cells, 44% in 23 slices), fewer cells cultured in the presence of taxol exhibited this behaviour (13/51 cells, 25% in 26 slices) (*Figure 5C*, *Figure 5—video 6*).

To test this requirement for microtubules during neuronal delamination further we additionally used a genetic approach. This involved mis-expression of Stathmin, which binds to soluble/free tubulin doublets (*Jourdain et al., 1997*) and so can be used to deplete soluble tubulin available for microtubule polymerisation (*Gavet et al., 1998*). Cells transfected with Stathmin-GFP and pCIG-Neurog2-NLS were monitored for 12–15 hr and delamination was quantified in cells poised to abscise. We found that few cells delaminated in the presence of Stathmin-GFP (3/11 cells, 27% in 6 slices, 5 experiments) (*Figure 5—video 7*), while many more cells underwent this step when only the vector control EGFP was expressed (10/17 cells, 59%, in 9 slices, 6 experiments) (*Figure 5—figure*

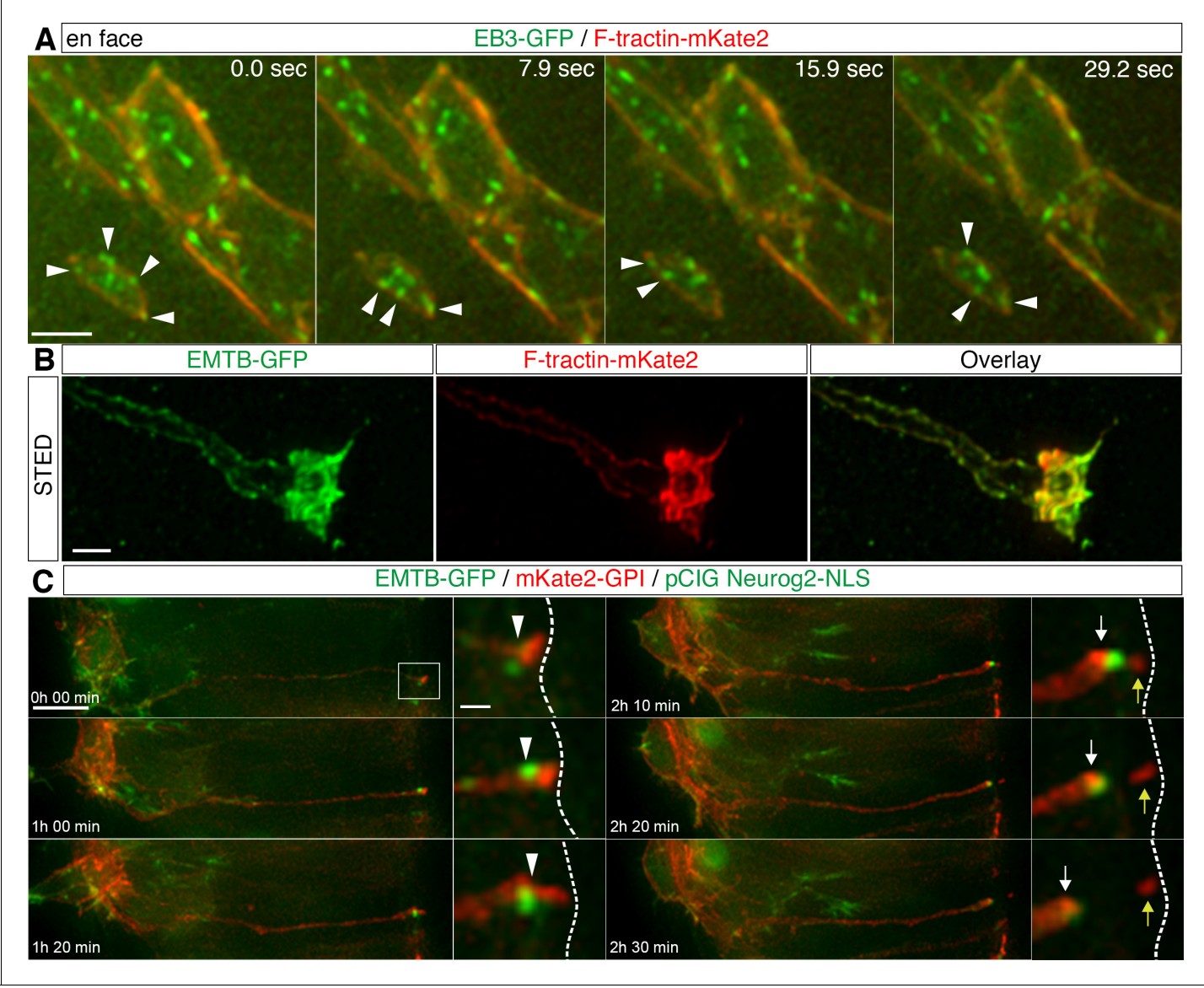

**Figure 4.** Apical cytoskeletal changes in delaminating cells. (A) In cells with small apical end-feet, EB3-GFP comets still radiate towards and become closely associated with the actin cable (white arrowheads). (B) STED image of a differentiating neuron end-foot mis-expressing EMTG-GFP (green) and F-tractin-mKate2 (red). (C) Time-lapse sequence of microtubule dynamics during apical abscission. Embryo neural tubes were electroporated with EMTB-GFP (green), pCIG-Neurog2 (nuclear, green) and mKate2-GPI (red). Abscission site (white arrowheads), withdrawing apical process (white arrows), abscised particle (yellow arrows) and apical side (white dashed line). Scale bars, (A) 2 μm, (B) 1 μm, (C) 10 μm, enlarged regions, 2 μm.

DOI: https://doi.org/10.7554/eLife.26215.017

The following videos are available for figure 4:

**Figure 4—video 1.** Time-lapse sequence of microtubule dynamics in cells with apical end-feet of reduced area; this video is related to *Figure 4A*.
DOI: https://doi.org/10.7554/eLife.26215.018

**Figure 4—video 2.** STED 3D reconstruction of apical end-foot of cell progressing through apical abscission; this video is related to *Figure 4B*.
DOI: https://doi.org/10.7554/eLife.26215.019

**Figure 4—video 3.** Time-lapse sequence of microtubule dynamics during apical abscission; this video is related to *Figure 4C*.
DOI: https://doi.org/10.7554/eLife.26215.020

supplement 3A and B) (*Figure 5—video 8*). These data indicate that microtubules and their turn-over and active growth are required for neuronal delamination.

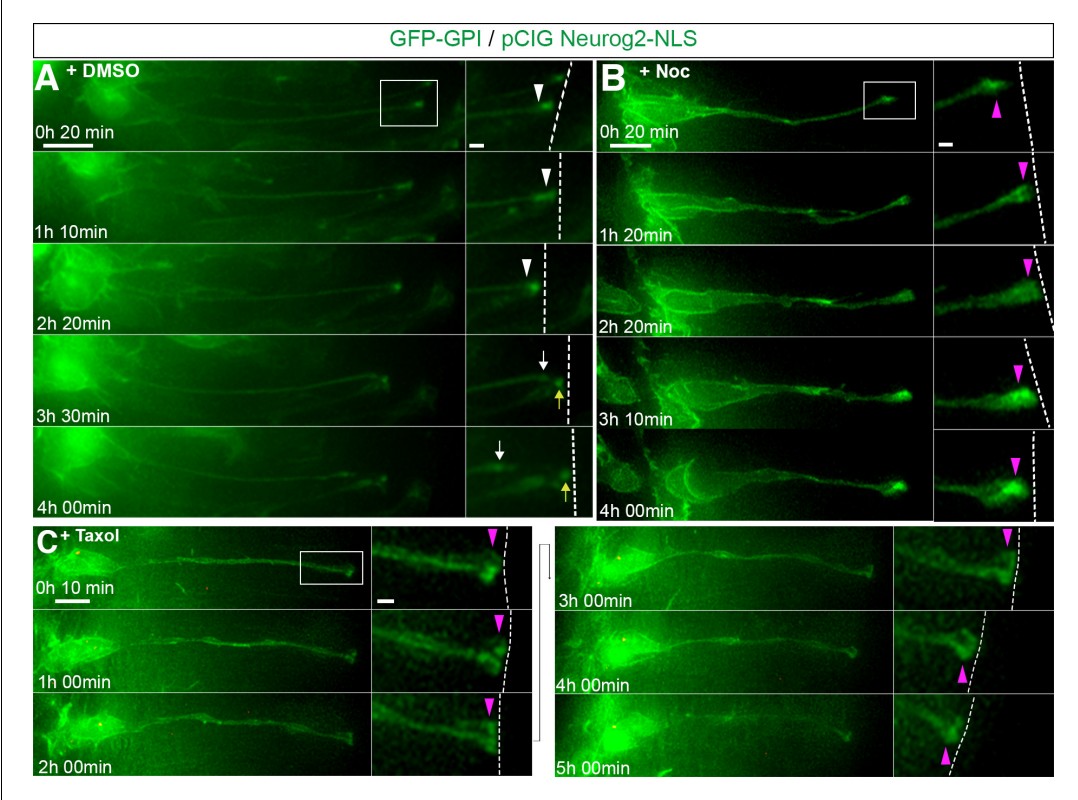

**Figure 5.** Apical abscission depends on dynamic microtubules. (**A**) Time-lapse sequence of cell imaged in medium containing DMSO vehicle control undergoing apical abscission. (**B**) Time-lapse sequence of cell imaged in medium containing nocodazole. (**C**) Time-lapse sequence of cell imaged in medium containing taxol. Embryo neural tubes were electroporated with GFP-GPI (cell membrane, green) and pCIG-Neurog2 (nucleus, green). Here and *Figure 5—figure supplement 5* : Apical end process (purple arrowhead), abscission site (white arrowheads), withdrawing apical process (white arrows), abscised particle (yellow arrows) and apical side (white dashed line). Scale bars: 10 μm; enlarged region, 2 μm.

DOI: https://doi.org/10.7554/eLife.26215.021

The following video and figure supplements are available for figure 5:

**Figure supplement 1.** Assessment of nocodazole treatment efficacy in neural tube slices.

DOI: https://doi.org/10.7554/eLife.26215.022

**Figure supplement 2.** Assessment of taxol treatment efficacy in neural tube slices.

DOI: https://doi.org/10.7554/eLife.26215.023

**Figure supplement 3.** Stathmin-GFP mis-expression results in reduced delamination.

DOI: https://doi.org/10.7554/eLife.26215.024

**Figure 5—video 1.** Time-lapse sequence of cells imaged in medium containing DMSO-vehicle control; this video is related to *Figure 5A*.

DOI: https://doi.org/10.7554/eLife.26215.025

**Figure 5—video 2.** Time-lapse sequence of cells imaged in medium containing nocodazole; this video is related to *Figure 5B*.

DOI: https://doi.org/10.7554/eLife.26215.026

**Figure 5—video 3.** Time-lapse sequence of cells imaged in medium containing nocodazole; this video relates to *Figure 5—figure supplement 1B*.

DOI: https://doi.org/10.7554/eLife.26215.027

**Figure 5—video 4.** Time-lapse sequence of cells imaged in medium containing DMSO vehicle control; this video relates to *Figure 5—figure supplement 1C*.

DOI: https://doi.org/10.7554/eLife.26215.028

**Figure 5—video 5.** Time-lapse sequence of cells imaged in medium containing taxol; this video relates to *Figure 5—figure supplement 2A*.

DOI: https://doi.org/10.7554/eLife.26215.029

**Figure 5—video 6.** Time-lapse sequence of cells imaged in medium containing taxol; this video is related to *Figure 5C*.

DOI: https://doi.org/10.7554/eLife.26215.030

**Figure 5—video 7.** Time-lapse sequence of cell dynamics following Stathmin-GFP mis-expression; this video is related to *Figure 5—figure supplement 3A*.

DOI: https://doi.org/10.7554/eLife.26215.031

*Figure 5 continued on next page*

*Figure 5 continued*

**Figure 5—video 8.** Time-lapse sequence of cell dynamics following pEGFP-N1 mis-expression; this video is related to *Figure 5—figure supplement 3B*.

DOI: https://doi.org/10.7554/eLife.26215.032

## Apical microtubule and actin conformational dynamics are inter-dependent in delaminating cells

To confirm that the association between microtubules and actin continues throughout abscission, we performed further live imaging of cells co-transfected with F-tractin-td-Tomato, EMTB-GFP and pCIG-Neurog2. We again observed that sub-apical actin and microtubules accumulated in and were closely associated at the abscission site and that this remained until final abscission, following which both actin and microtubules were rapidly depleted from the cell-process tip (*Figure 6A*, *Figure 6—video 1*, 12 cells in 6 slices from 5 embryos).

To investigate the potential regulatory interactions between actin and microtubules, we next used taxol to stabilise microtubules in cells expressing mKate2-GPI, pCIG-Neurog2 and EMTB-GFP that were poised to delaminate. This confirmed cessation of EMTB-GFP accumulation and subsequent failure to detach from the apical surface. These EMTB-GFP dynamics were then quantified by measuring GFP fluorescence intensity at the sub-apical poles of these cells following exposure to this drug (*Figure 6B*, quantified in B'' grey dashed line, *Figure 6—video 2*, 12 cells in 10 slices). These clearly contrasted with control cells imaged in medium containing only DMSO which displayed normal accumulation and subsequent loss of EMTB-GFP during abscission (*Figure 6B'*, quantified in B'' (black dashed line), *Figure 6—video 3*, 12 cells in 11 slices). We then monitored overall actin dynamics in cells expressing GFP-GPI, pCIG-Neurog2 and F-tractin-mKate2 that were poised to delaminate. We observed that while some cells in taxol-treated slices exhibited sub-apical constriction as judged by local cell shape change (*Figure 6C* white arrowheads,16/34 cells in 21 slices), this dynamic of sub-apical actin accumulation and subsequent loss ceased as indicated by fluorescence intensity measurements and such cells remained attached at the ventricular surface (*Figure 6C*, quantified in C'' (grey dashed line), *Figure 6—video 4*, 20 cells in 16 slices). In contrast, actin intensity in cells imaged in medium containing only DMSO increased at the abscission site and was then rapidly lost from the withdrawing cell-process (*Figure 6C'*, quantified in C'' (black dashed line), *Figure 6—video 5*, 12 cells in 11 slices), consistent with our previous report of actin dynamics during this process (*Das and Storey, 2014*).

We then carried out the converse experiment, in which cells expressing mKate2-GPI, pCIG-Neurog2 and EMTB-GFP that were poised to delaminate were cultured in medium containing 20 µM ML-7 to inhibit acto-myosin constriction (*Saitoh et al., 1987*). We observed that the majority of the EMTB-GFP electroporated cells were now unable to initiate sub-apical constrictions (19/25 cells in 18 slices) and progress through to abscission, and that sub-apical EMTB-GFP labelling no longer exhibited its characteristic pattern of accumulation followed by loss. This was confirmed by measuring sub-apical GFP fluorescence intensities in these cells (*Figure 6D* and quantified in D'' (grey dashed line), *Figure 6—video 6*, 16 cells in 13 slices). This profile contrasted with control cells, in which EMTB-GFP accumulated and was subsequently lost following abscission (*Figure 6D'* and quantified in D'' (black dashed line), *Figure 6—video 7*, 10 cells in 10 slices). These findings indicate that microtubule and actin conformational dynamics are inter-dependent; loss of actively growing microtubules blocked stable accumulation of actin at the presumptive abscission site and loss of acto-myosin activity abolished enrichment of microtubules at this location. Consistent with this inter-dependence, inhibition of either acto-myosin (*Das and Storey, 2014*) or microtubule activity (*Figure 5C*) reduced the incidence of neuronal delamination.

## The actin and microtubule cross-linking protein drebrin is required for neuronal delamination

A number of proteins have been proposed to link actin and microtubules (*Coles and Bradke, 2015*). These include Drebrin, which was initially identified as an actin-binding protein (*Ishikawa et al., 1994*) and later shown to interact with the +TIP protein EB3 (*Geraldo et al., 2008*). To address whether Drebrin is a candidate mediator for actin-microtubule interaction during neuronal

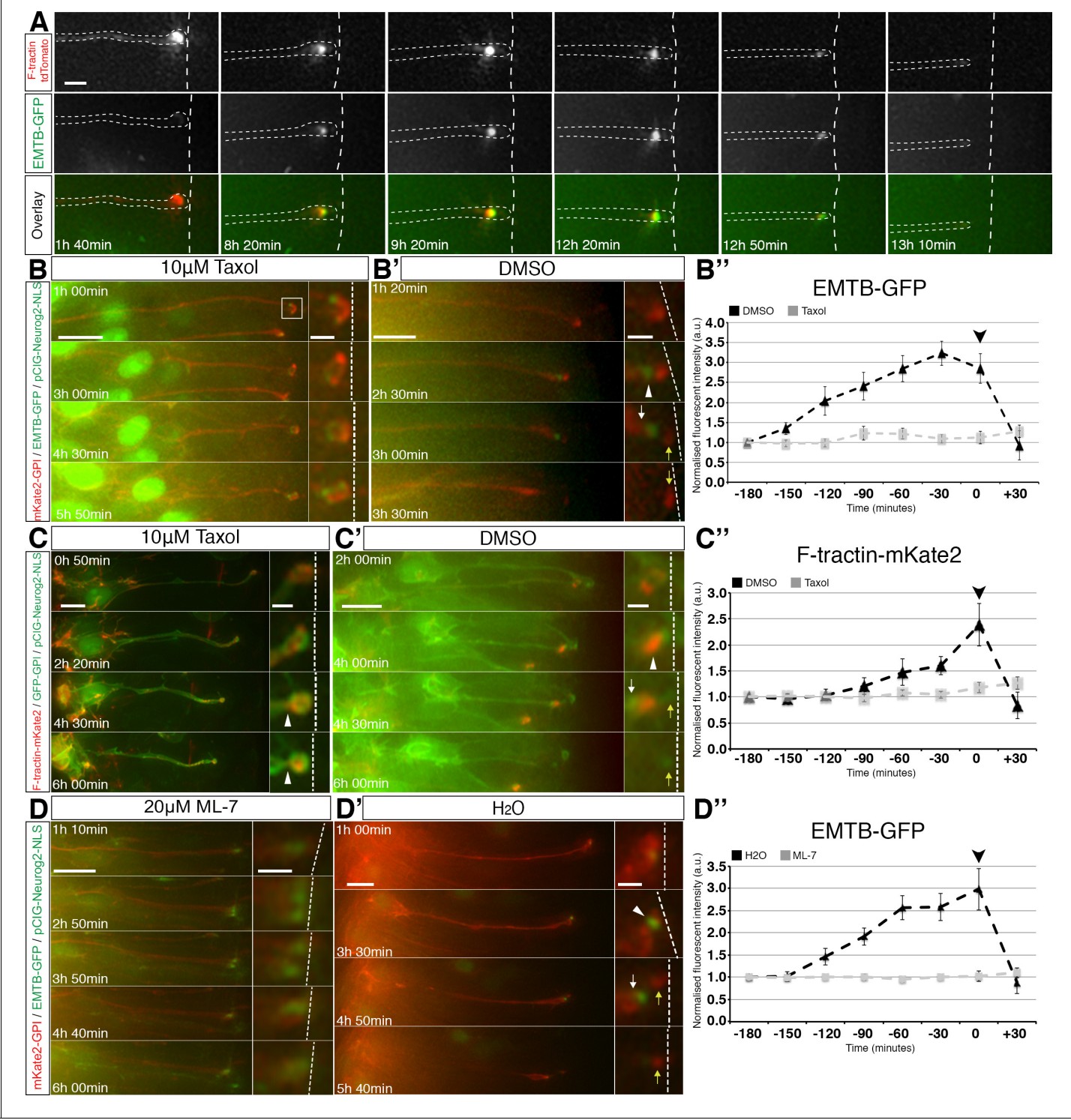

**Figure 6.** Coordination of sub-apical actin and microtubule dynamics. (A) Live imaging of sub-apical actin (F-tractin td-Tomato) and microtubule (EMTB-GFP) dynamics during apical abscission. (B–B', C–C', D–D'). Time-lapse sequences of neural tube in embryo slices electroporated with EMTB-GFP/pCIG-Neurog2/mKate2 GPI or F-tractin-mKate2/pCIG-Neurog2/GFP GPI and treated with taxol (B, C) or ML-7 (D) or control vehicle (B', C', D'). Abscission site (white arrowheads), withdrawing apical process (white arrows), abscised particle (yellow arrows) and apical side (white dashed line) (B'', C'', D'') Line graphs of normalised fluorescence intensities of EMTB-GFP or F-tractin-mKate2 dynamics in taxol or ML-7 (grey dashed line) and their control vehicles (black dashed line), quantified for 3 hr 30 min at 30 min intervals. EMTB-GFP dynamics are significantly affected by the taxol and ML-7 treatment (2-way ANOVA, p<0.001 for each of the treatments, error bars = SEM). F-tractin-mKate2 dynamics are significantly affected by ML-7

*Figure 6 continued on next page*

*Figure 6 continued*

treatment (2-way ANOVA, p=0.002, error bars = SEM). Black arrowhead is abscission point for controls. Scale bars, (A) 2 μm, (B–B') (C–C') (D–D') 10 μm; enlarged regions, 2 μm.Figure

DOI: https://doi.org/10.7554/eLife.26215.033

The following video and source data are available for figure 6:

**Source data 1.** Quantification of EMTB-GFP and F-tractin-mKate2 fluorescence.

DOI: https://doi.org/10.7554/eLife.26215.034

**Figure 6—video 1.** Time-lapse sequence of actin and microtubule dynamics during apical abscission; this video is related to *Figure 6A*.

DOI: https://doi.org/10.7554/eLife.26215.035

**Figure 6—video 2.** Time-lapse sequence of microtubule dynamics in cells imaged in medium containing taxol; this video is related to *Figure 6B*.

DOI: https://doi.org/10.7554/eLife.26215.036

**Figure 6—video 3.** Time-lapse sequence of microtubule dynamics in cells imaged in medium containing DMSO vehicle control; this video is related to *Figure 6B'*.

DOI: https://doi.org/10.7554/eLife.26215.037

**Figure 6—video 4.** Time-lapse sequence of actin dynamics in cells imaged in medium containing taxol; this video is related to *Figure 6C*.

DOI: https://doi.org/10.7554/eLife.26215.038

**Figure 6—video 5.** Time-lapse sequence of actin dynamics in cells imaged in medium containing DMSO vehicle control; this video is related to *Figure 6C'*.

DOI: https://doi.org/10.7554/eLife.26215.039

**Figure 6—video 6.** Time-lapse sequence of microtubule dynamics in cells imaged in medium containing ML-7; this video is related to *Figure 6D*.

DOI: https://doi.org/10.7554/eLife.26215.040

**Figure 6—video 7.** Time-lapse sequence of microtubule dynamics in cells imaged in H2O vehicle control; this video is related to *Figure 6D'*.

DOI: https://doi.org/10.7554/eLife.26215.041

delamination we first assessed localisation of endogenous protein using IHC in transverse sections of the neural tube (*Figure 7A*). We found widespread cytoplasmic localisation of endogenous Drebrin, including in the apical end-foot (*Figure 7A–A'''*, 3 sections from each of 4 embryos). To look more closely at Drebrin localisation in end-feet we mis-expressed Drebrin-mCherry and EMTB-GFP and stained for actin in individual cells (*Figure 7B–B''*, 18 cells, 6 embryos). This analysis confirmed cytoplasmic localisation but also revealed co-localisation with the actin belt and the apical EMTB-GFP-labelled microtubules, quantified by measuring fluorescence intensities across the actin cable in a subset of cells (*Figure 7C*, 7 cells, 3 embryos, see Materials and methods). Similar co-localisation of Drebrin-YFP and actin was also apparent in en face images (*Figures 7D*, 5 explants from 5 embryos). These localisation studies support the possibility that Drebrin is involved in the coordination of actin and microtubule dynamics during neuronal delamination

To test the requirement for Drebrin in this process, we next mis-expressed a Drebrin short-hairpin (Sh) construct (*Dun et al., 2012*) in the developing neural tube along with Neurog2. This led to a marked reduction in the number of delaminating cells (4/27 cells in 12–15 hr, 15% in 9 slices) compared to the scrambled GFP control (7/10 cells in 12–15 hr, 70% in 7 slices) (*Figure 7—figure supplement 1*, *Figure 7—videos 1* and *2*). This requirement for Drebrin during neuronal delamination is consistent with a role for this protein in regulating cytoskeletal dynamics during this process and supports the possibility that Drebrin acts here as a link between actin-microtubules.

## The centrosome translocates through a tunnel-like actin-microtubule configuration and this relies on active acto-myosin and microtubules

Apical abscission is characterised by dis-assembly of the centrosome-primary cilium complex which is followed by a basal translocation of the centrosome and so its retention in the withdrawing cell-process (*Das and Storey, 2014*). To investigate the relationship between this translocation and the sub-apical constriction, neural tube cells were transfected with GFP-GPI, pCIG-Neurog2 and PACT-TagRFP, which labels centrosomes and cells were then subjected to live imaging. We observed that differentiating neurons first constricted their sub-apical membranes and that this was then strikingly followed by basal translocation of the centrosome. This event therefore takes place late in the delamination process; indeed in some cells this movement was visible within a thinned membrane bridge between the withdrawing cell-process and the abscising particle (*Figure 8A*, *Figure 8—video 1*, 10 cells, 9 slices in 9 embryos). Monitoring centrosome translocation in cells expressing

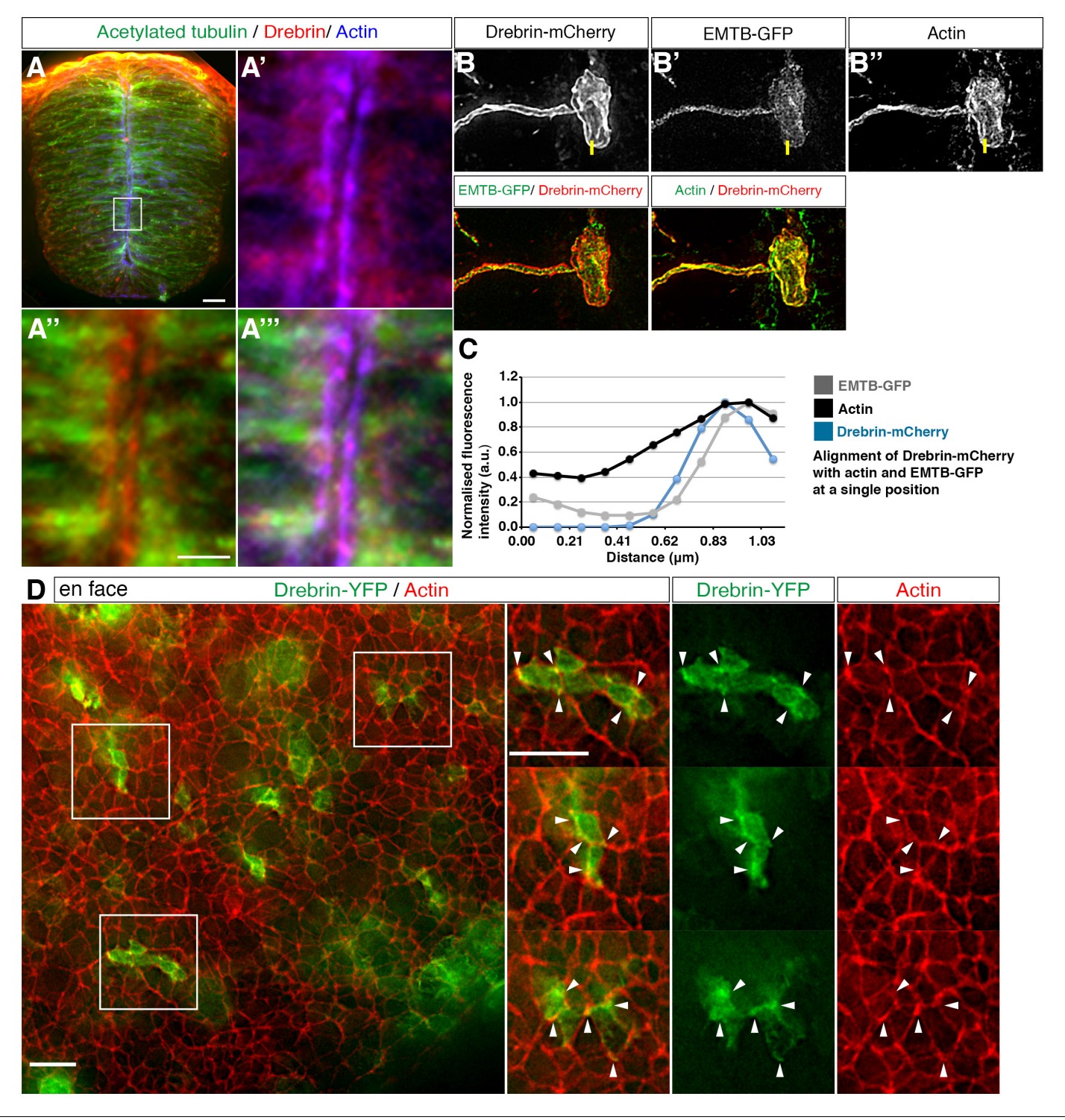

**Figure 7.** Drebrin localisation in the neural tube. (**A**) Representative image of HH17-18 chick embryo neural tube labelled with antibodies to detect drebrin and acetylated α-tubulin and stained with phalloidin. Magnified boxed region shown in **A'-A'''**. End-foot of a neuroepithelial cell mis-expressing (**B**) Drebrin-mCherry and (**B'**) EMTB-GFP and stained with (**B''**) phalloidin. (**C**) Representative line graphs of normalised fluorescence intensity at the level of the actin cable (**B–B''**). (**D**) En face imaging of neuroepithelial end-feet electroporated with Drebrin-YFP and stained for phalloidin. Boxed areas are magnified. White arrowheads indicate Drebrin-YFP and phalloidin co-localisation. Scale bars, (**A**) 20 μm and boxed region 5 μm, (**B**) 2 μm, (**C**) 10 μm.

DOI: https://doi.org/10.7554/eLife.26215.042

The following video, source data, and figure supplement are available for figure 7:

*Figure 7 continued on next page*

*Figure 7 continued*

**Source data 1.** Drebrin-mCherry / Actin / EMTB-GFP alignment.
DOI: https://doi.org/10.7554/eLife.26215.044
**Figure supplement 1.** Drebrin knockdown reduces the incidence of delamination.
DOI: https://doi.org/10.7554/eLife.26215.043
**Figure 7—video 1.** Time-lapse sequence of cell behaviour following Drebrin knockdow; this video is related to *Figure 7—figure supplement 1A*.
DOI: https://doi.org/10.7554/eLife.26215.045
**Figure 7—video 2.** Time-lapse sequence of cell behaviour following scrambled shRNA construct expression; this video is related to *Figure 7—figure supplement 1B*.
DOI: https://doi.org/10.7554/eLife.26215.046

PACT-TagRFP and EMTB-GFP further revealed that the translocating centrosome moves basally before the resolution of the sub-apical microtubules (*Figure 8B*, *Figure 8—video 2*, 8 cells in 8 slices and see *Figure 4B*). This suggests that it passes through the sub-apical actin/microtubule tunnel-like configuration that we observed in cells poised to delaminate (*Figure 4B*, *Figure 4—video 2*). To investigate this possibility further, we measured the diameter of the ring formed by rim microtubules visualised with acetylated alpha tubulin (17 cells in 2 explants from 2 embryos, *Figure 8—figure supplement 1*). This gave an average diameter of $0.89 \pm 0.18$ μm with an average centrosome diameter measured with IFT88, at the base of the ciliary membrane, of $0.32 \pm 0.06$ μm (36 cells in 2 explants from 2 embryos). However, the latter only identifies the ciliary axoneme and centrosome (*Robert et al., 2007*) and so may under-estimate the full extent of the centrosomal material. Measurement of centrosomal γ-tubulin (which includes peri-centriolar material) revealed an average size of $0.98 \pm 0.12$ (21 cells, data not shown), consistent with centrosome size of $0.82 \pm 0.17$ μm in other contexts (*Fu and Glover, 2012*). These data therefore support the possibility that the centrosome moves through a tunnel-like cytoskeletal configuration formed by apical microtubules and the constricting actin cable.

These observations raised the further possibility that sub-apical constriction, which depends on acto-myosin activity, is required for subsequent centrosome translocation. To test this, we observed cells in slices transfected with GFP-GPI, pCIG-Neurog2 and PACT-TagRFP that were cultured in medium containing ML-7, to block acto-myosin constriction. In such conditions, few cells delaminated and exhibited sub-apical constrictions or centrosome translocation within 6 hr (*Figure 8C*, *Figure 8—video 3*, 5/31 cells in 9 slices). To determine whether centrosome translocation also required active microtubules, slices transfected with the same constructs were exposed to 10 μM taxol, and again fewer cells exhibited centrosome translocation and abscised within 6 hr (*Figure 8D*, *Figure 8—video 4*, 4/24 cells in 12 slices) compared with DMSO control conditions (*Figure 8E*, *Figure 8—video 5*, 9/26 cells in 14 slices). These experiments indicate that centrosome translocation and hence its retention in the newborn neuron depends on both microtubule turnover and acto-myosin constriction.

## Centrosome nucleated microtubules are required for delamination

The centrosome is important for subsequent morphogenesis of the newborn neuron, but it is unclear whether it is also involved in the delamination process. Indeed, while the centrosome has been implicated in the final stages of cytokinetic abscission (*Piel et al., 2001*) it is also possible that ablating this organelle might hasten loss of microtubule-actin/cadherin interactions and so trigger delamination.

To investigate the involvement of the centrosome in this process, this structure was disrupted using chromophore assisted light inactivation (CALI) mediated by the phototoxic fluorescent protein KillerRed (*Bulina et al., 2006*) linked to the pericentrin derived PACT domain. To verify centrosome disruption using this approach, cells were first transfected with PACT-KillerRed and PACT-YFP. Following irradiation with green light, we observed photo-bleaching of the PACT-KillerRed labelling and a corresponding reduction in PACT-YFP labelling (*Figure 9A*, 5/5 cells in 5 slices), indicating that photoactivation of KillerRed compromised neighbouring centrosomal protein complexes. Conversely, cells transfected with PACT-TagRFP and PACT-YFP and exposed to the same regime did not display reduced YFP labelling (*Figure 9B*, *Figure 9—video 1*, 25/25 cells in 4 slices from 4 embryos), supporting the conclusion that CALI mediated by PACT-KillerRed targeted centrosomal

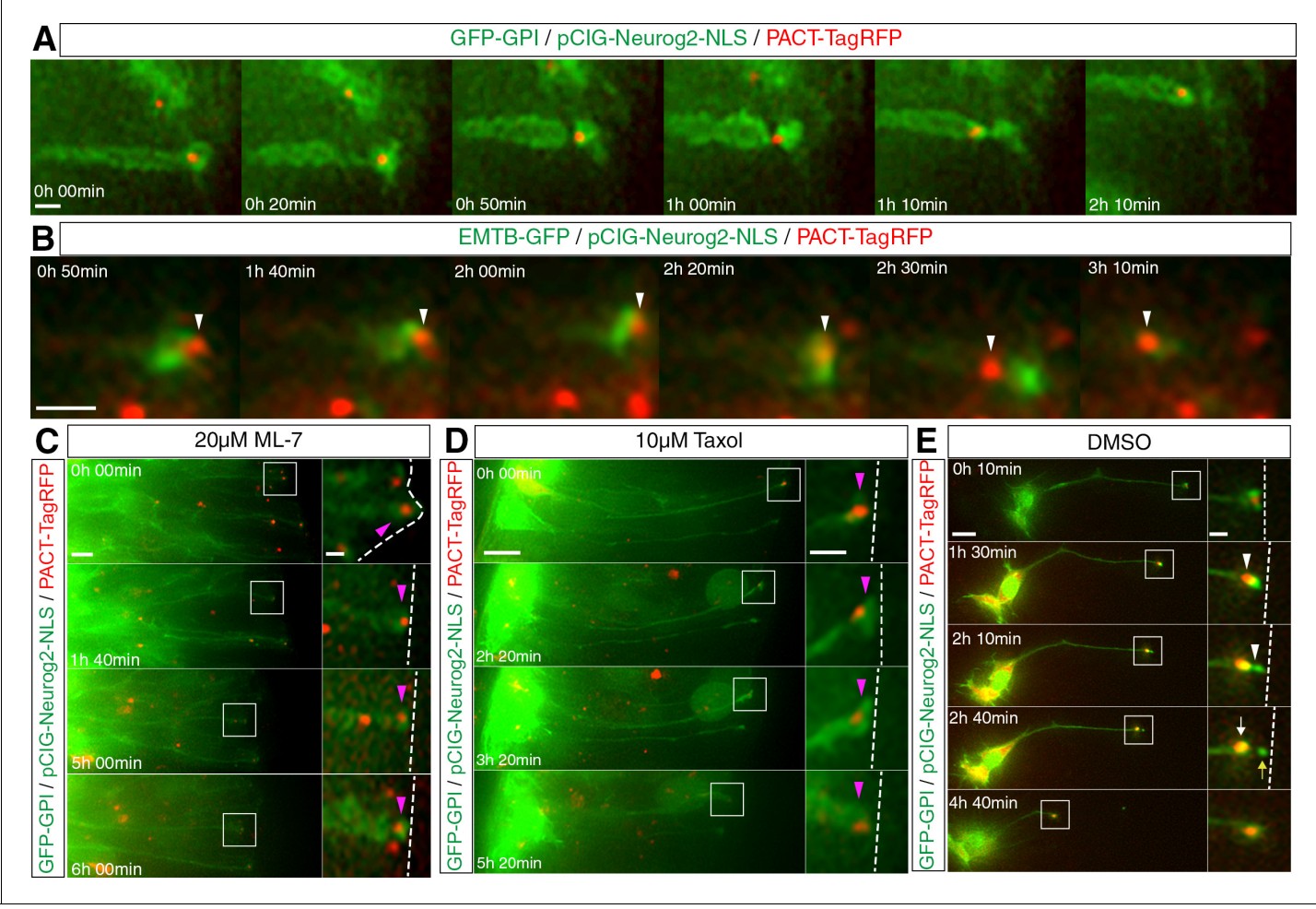

**Figure 8.** Centrosome translocation during apical abscission depends on actin and microtubule dynamics. (A) The centrosome (labelled with PACT-TagRFP, red) undergoes a basal translocation through a thinned region of membrane (labelled with GFP-GPI, green). (B) The centrosome (labelled with PACT-TagRFP, red) translocates through a condensed microtubule tunnel-like configuration (labelled with EMTB-GFP, green). (C–E) Time-lapse sequences of centrosome dynamics in cells imaged in medium containing ML-7 (C), taxol (D) or DMSO control (E). Embryo neural tubes electroporated with GFP-GPI/pCIG-Neurog2/PACT TagRFP. Apical end process (purple arrowhead), abscission site (white arrowheads), withdrawing apical process (white arrows), abscised particle (yellow arrows) and apical side (white dashed line). Scale bars, (A) (B) 2 μm, (C) (D) (E) 10 μm; enlarged regions, 2 μm.

DOI: https://doi.org/10.7554/eLife.26215.047

The following video and figure supplement are available for figure 8:

**Figure supplement 1.** Measurement of apical microtubule rim and centrosome diameter.
DOI: https://doi.org/10.7554/eLife.26215.048

**Figure 8—video 1.** Time-lapse sequence of centrosome undergoing basal translocation; this video is related to *Figure 8A*.
DOI: https://doi.org/10.7554/eLife.26215.049

**Figure 8—video 2.** Time-lapse sequence of centrosome translocation through a condensed microtubule tunnel-like configuration; this video is related to *Figure 8B*.
DOI: https://doi.org/10.7554/eLife.26215.050

**Figure 8—video 3.** Time-lapse sequence of centrosome dynamics in cells imaged in medium containing ML-7; this video is related to *Figure 8C*.
DOI: https://doi.org/10.7554/eLife.26215.051

**Figure 8—video 4.** Time-lapse sequence of centrosome dynamics in cells imaged in medium containing taxol; this video is related to *Figure 8D*.
DOI: https://doi.org/10.7554/eLife.26215.052

**Figure 8—video 5.** Time-lapse sequence of centrosome dynamics in cells imaged in medium containing DMSO-vehicle control; this video is related to *Figure 8E*.
DOI: https://doi.org/10.7554/eLife.26215.053

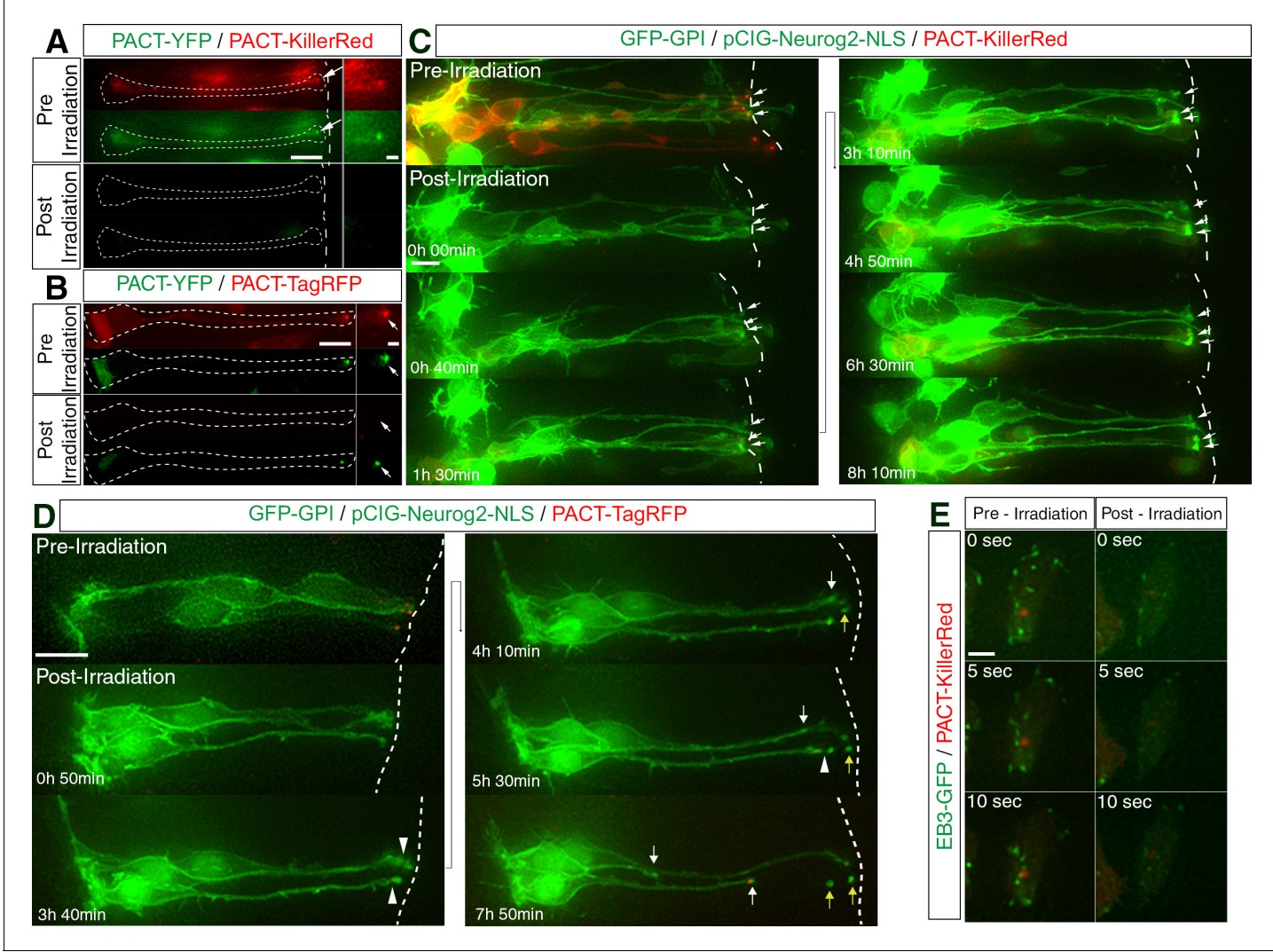

**Figure 9.** Compromised microtubule nucleating potential of the centrosome blocks delamination from the apical surface. (A) Green light irradiation-mediated photobleaching of PACT-KillerRed, which localises to the centrosome, is accompanied by a corresponding depletion of PACT-YFP fluorescence. Arrows point to the centrosome, (B) Photobleaching of PACT-TagRFP following green light irradiation does not result in a corresponding reduction of PACT-YFP fluorescence. Arrows point to the centrosome. (C) Time-lapse sequence of neural progenitors following CALI. Cells poised to differentiate remain attached to the apical surface. Cells were electroporated with pCIG-Neurog2, GFP-GPI and PACT-KillerRed. White arrows point to the apical tips of cells that have been subjected to CALI. (D) Time-lapse sequence of neural progenitors following the imaging regime used for CALI. Two out of three cells underwent apical abscission during the 8 hr post-irradiation imaging period. Cells were electroporated with pCIG-Neurog2, GFP-GPI and PACT-TagRFP in place of the PACT-KillerRed construct. Abscission site (white arrowheads), withdrawing apical process (white arrows), abscised particle (yellow arrows) and apical side (white dashed line). (E) Reduction in the microtubule nucleation potential (48% reduction) of the centrosome, 3 hr post-CALI. Stills of EB3-GFP comets pre- and post-irradiation of a single end-foot (en face). Scale bars, (A–D) 10 μm; enlarged regions, 2 μm, (E) 2 μm.

DOI: https://doi.org/10.7554/eLife.26215.054

The following videos are available for figure 9:

**Figure 9—video 1.** Time-lapse sequence of cells following green-light irradiation; this video is related to *Figure 9B*.
DOI: https://doi.org/10.7554/eLife.26215.055

**Figure 9—video 2.** Time-lapse sequence of cell behaviour following CALI; this video is related to *Figure 9C*.
DOI: https://doi.org/10.7554/eLife.26215.056

**Figure 9—video 3.** Time-lapse sequence of cell behaviour following the CALI green light irradiation; this video is related to *Figure 9D*.
DOI: https://doi.org/10.7554/eLife.26215.057

proteins. To assess the functional significance of this manipulation we then carried out this CALI experiment and monitored PACT-KillerRed and production of EB3-GFP comets. This revealed a dramatic reduction in the number of comets (assessed at a 3 hr time point post-CALI) (22 cells in 7 explants from 7 embryos, *Figure 9E*) indicating that this regime significantly compromised centrosome-mediated microtubule nucleation. We then performed CALI on cells transfected with PACT-KillerRed, GFP-GPI and pCIG-Neurog2 that were poised to delaminate. This resulted in fewer cells detaching from the ventricular surface during the subsequent 8 hr imaging period (3/12 cells, 25%, in 5 slices *Figure 9C*, *Figure 9—video 2*), compared with control PACT-TagRFP transfected cells (15/25 cells, 60%, in 10 slices, *Figure 9D*, *Figure 9—video 3*). This suggests that centrosome-mediated microtubule nucleation is required for delamination.

## Discussion

In this study, we elucidate cytoskeletal architecture and regulatory relationships between actin and microtubules that maintain the neuroepithelial apical end-foot and how these alter to direct neuronal delamination. We uncover a conserved wheel-like microtubule organisation, composed of rim and radial microtubules nucleated by the centrosome, which spans the apical end-foot and aligns with the actin cable and linked AJs. We show that apical actin maintains these microtubules, which are in turn required for maintenance of AJs and that apical actin serves to define end-foot dimensions. This apical cytoskeleton then changes dramatically in newborn neurons as they undergo apical abscission and delaminate following downregulation of N-Cadherin. The apical microtubules condense led by the constricting actin cable and together these form a tunnel-like configuration through which the centrosome then transits as it moves basally. We demonstrate that this enrichment of microtubules at the abscission site depends on acto-myosin activity and that dynamic microtubules are in turn required for effective acto-myosin constriction. We additionally identify the actin-microtubule cross-linking protein Drebrin as a potential coordinator of microtubule and actin dynamics and demonstrate its requirement for neuronal delamination. Furthermore, inter-dependent actin and microtubule dynamics were required for centrosome translocation and subsequent cell detachment. Indeed, compromise of centrosome microtubule-nucleating capacity decreased the incidence of delamination, indicating that this organelle is a critical promoter of new microtubules mediating this step. These data demonstrate that neuronal delamination is an active process; it is not sufficient to downregulate N-cadherin, nucleation of apical microtubules and inter-dependent microtubule and actin dynamics are needed to drive this process and to retain the centrosome in the newborn neuron.

### Neuroepithelial apical end-foot architecture relies on actin and microtubule maintenance of adherens junctions

One of the major challenges in neural development as well as cell biology is to elucidate the mechanisms regulating cytoskeletal interactions that direct neuroepithelial integrity and neuronal morphology. We provide evidence here for a microtubule wheel-like organization nucleated by the centrosome of the primary cilium in neuroepithelial apical end-feet and for conservation of this configuration across species and regions of the central nervous system. A similar wheel-like arrangement of microtubules has been observed in kidney epithelial (MDCK) cells in vitro and cochlear epithelial cells (*Bellett et al., 2009*). Here centrosomal microtubules were orientated with plus-ends towards the AJ (*Bellett et al., 2009*) and by tracking the trajectories of EB3-GFP comets we observed a similar configuration in neuroepithelial cells. One explanation for this structure is the recruitment of microtubule plus-ends by AJs/cell cell contact, which has been demonstrated in several epithelial cell lines in vitro (*Stehbens et al., 2006*; *Waterman-Storer et al., 2000*). However, in myoblasts, microtubules are directed towards cell contacts by their plus-ends, and here they are then locally repelled at N-cadherin adhesion sites (*Plestant et al., 2014*). This indicates that AJ capture of microtubules is context dependent; indeed this can involve association with minus- rather than plus-ends (*Meng et al., 2008*) and that other mechanisms might also account for plus-end growth towards the cell periphery. We show here that in neuroepithelial cells microtubule wheel-like 'rim' microtubules interface with the actin cable and that this configuration is generated by dynamic centrosome generated microtubules that bend and grow along the actin cable. This may reflect bio-physical properties of microtubules when they encounter the epithelial cell periphery (*Gomez et al., 2016*) and/or regulation by proteins transported by microtubules (*Mata and Nurse, 1997*), but it also suggests that

interaction between these two cytoskeletal components influences overall microtubule conformation. Our data support such a regulatory relationship, demonstrating microtubule depletion in the apical end-foot following inhibition of actin polymerisation and increased accumulation of actin within the cell following depletion of microtubules; indicating that microtubules regulate actin localisation, although levels of actin at the adhesion belt were unaffected in the timeframe of our assay. Such interactions may be mediated directly by proteins that bind actin and microtubules, these may include formins, IQGAP, dynein/dynactin complex and unconventional myosins as well as Drebrin (*Bazellières et al., 2012*; *Brown, 1999*; *Geraldo et al., 2008*; *Goode et al., 2000*; *Merriam et al., 2013*; *Rodriguez et al., 2003*; *Trivedi et al., 2017*).

We demonstrate here that Drebrin is localised in apical end-feet of neuroepithelial cells in a distribution similar to that in apical intestinal epithelia (*Bazellières et al., 2012*) that includes the sub-apical actin cable, which we show is also aligned with apical microtubules. Drebrin is therefore in a position to link and so coordinate changes in the acto-myosin cytoskeleton and microtubules during neuronal delamination. Furthermore, *Drebrin* knock-down clearly indicated that this protein is required for neuronal delamination. Experiments should now be focused on elucidating Drebrin dynamics during this process in relation to those of actin and microtubules. In particular, it will be important to establish whether Drebrin serves to direct EB3 comets emerging from the centrosome to actin cable and so create the interface between apical microtubules and sub-apical actin, much as observed during neuronal cell nucleokinesis and migration movements (*Trivedi et al., 2017*). Drebrin binding of the AJ protein Afadin (*Rehm et al., 2013*) also raises the interesting possibility that changes in Drebrin localisation as these junctions disassemble, underpins coordinated condensation of the actin and microtubule cytoskeleton during delamination.

In other cellular contexts, emphasis has been placed on microtubule regulation of AJs. There is evidence that microtubules promote accumulation of E-cadherin at epithelial cell-cell contacts (*Stehbens et al., 2006*; *Waterman-Storer et al., 2000*), but this did not reflect a role in conveying E-cadherin to the cell surface (*Stehbens et al., 2006*). However, these researchers demonstrated a requirement for microtubules for myosin phosphorylation at sites of E-cadherin accumulation in MCF7 cells and so linked microtubules to actin-mediated organisation of AJs. In contrast, N-cadherin transport to the cell membrane requires the microtubule kinesin based motor in a range of cell types (*Mary et al., 2002*; *Teng et al., 2005*) and neuroepithelial cells in mice mutant for the KIF3 motor complex protein KAP3, lack membrane localised N-Cadherin (*Teng et al., 2005*). Our data demonstrate that within an hour of microtubule depletion N-Cadherin levels drop dramatically at AJs, consistent with microtubule transport of N-cadherin in the neuroepithelial end-foot.

Importantly, actin is required to maintain these apical microtubules and both actin and microtubules maintain the AJs, so actin may act directly and/or indirectly to promote these junctions. Unlike nocodazole treatment, acute inhibition of actin filament assembly reduced actin levels at the adhesion belt and resulted in a smaller end-foot size and so indicated that it is the actin cable that determines apical end-foot dimensions.

## Neuronal delamination is driven by acto-myosin constriction and dynamic microtubules

This delicately balanced apical cytoskeletal architecture changes dramatically as newborn neurons delaminate from the neuroepithelium. This involves the process of apical abscission, which takes place following N-cadherin downregulation (*Das and Storey, 2014*; *Rousso et al., 2012*). We show here that this includes enrichment of microtubules as well as actin in a composite tunnel-like configuration at the presumptive abscission site. It is interesting that blocking microtubule growth with taxol, while not abolishing acto-myosin contractility, interferes with stable accumulation of actin and that this correlates with reduced cell delamination. This regulatory relationship appears similar to that of the central spindle during cytokinesis, which specifies assembly of the acto-myosin ring by delivering the small GTPase RhoA to the equatorial cortex, that in turn triggers local actin polymerisation and acto-myosin contractility (*Eggert et al., 2006*; *Piekny et al., 2005*). This relationship is also consistent with failure to disassemble AJs and impaired acto-myosin constriction in calcium-free conditions (which disrupt trans-cadherin dimers) in renal and intestinal cells treated with taxol in vitro, (*Ivanov et al., 2006*); which additionally suggests a further role for active microtubules in AJ disassembly.

Importantly, downregulation of N-Cadherin during neuronal delamination involves not simply transcriptional repression downstream of the neurogenesis transcription factor cascade (*Rousso et al., 2012*), but also mechanism(s) that remove N-Cadherin protein, as plasmid driven N-Cadherin is attenuated by such proneural gene activity (*Das and Storey, 2014*). One possibility is that rearrangement of apical microtubules during apical abscission may reduce microtubule-AJ association and so further facilitate loss of N-cadherin protein. This may additionally involve regulation of endocytosis/cadherin turnover and there is evidence that actin can also influence this process (*Cavey and Lecuit, 2009*; *Georgiou et al., 2008*; *Ivanov et al., 2004*; *Izumi et al., 2004*; *West and Harris, 2016*). For example, in a cell free assay trans-acting E-Cadherin activates the actin Rac1/Cdc42/IQGAP1 pathway that inhibits E-Cadherin endocytosis and so maintains AJs (*Izumi et al., 2004*); when such cell-cell interactions are lost then cadherin endocytosis increases. This mechanism is consistent with the phenotype of *Cdc42* deletion in the developing mouse cortex, which leads to loss of AJs and mis-localisation of neuroepithelial cells away from the ventricular/apical surface (*Cappello et al., 2006*) and with the involvement of heterotopia-associated genes FilaminA and ARFGEF2/BIG2 in endocytosis (*Sheen, 2014*).

In previous work, we established that acto-myosin constriction was required for apical abscission and here we show that inhibition of acto-myosin activity with ML-7 also blocks accumulation of microtubules at the presumptive abscission site. Together with the requirement for microtubules for stable actin accumulation, these findings suggest that active actin is upstream of microtubule conformational change during this process and that these microtubules then act back to promote effective acto-myosin constriction. Importantly, these data demonstrate that microtubules and actin continue to influence each other even when N-Cadherin/AJs are disassembled in a delaminating cell, further supporting involvement of cross-linking proteins which directly coordinate these cytoskeletal components. An intriguing possibility is that microtubules act here during delamination to augment myosin II phosphorylation, as reported at cell-cell contacts in MCF7 cells in vitro (*Stehbens et al., 2006*).

## Centrosome retention is linked to the abscission mechanism

The continued generation of radial comets from the centrosome in cells with small apical end-feet suggests that microtubule nucleation persists as the actin cable constricts and that this may result in formation of the microtubule/actin tunnel-like configuration through which the centrosome eventually passes. This is supported by our finding that both acto-myosin contractility and microtubule turnover are required for centrosome translocation. Furthermore, by specifically compromising centrosome-mediated microtubule nucleation using targeted CALI, we demonstrate that delamination requires centrosome generated microtubules. To our surprise, we further found that centrosome translocation takes place late in the abscission process, in highly constricted cells. Together these findings suggest a mechanism which places the centrosome at the centre of the abscission process and its own retention during neuronal delamination (*Figure 10*).

This sequence of events has some similarity to that taking place during cytokinesis observed in Hela cells (*Piel et al., 2000*; *Piel et al., 2001*); here, following cleavage furrow and midbody formation, movement of the mother centriole into the midbody bridge triggers release of central spindle microtubules, while disassembly of the actin ring and plasma membrane scission take place after it moves away (*Piel et al., 2000*). Furthermore, experiments which compromise the centrosome inhibited final cytokinetic abscission (*Piel et al., 2001*) or, in our experiments, neuronal delamination and this suggests that the centrosome provides molecular cues that prompt common final abscission steps. A critical function for the centrosome in neuronal delamination predicts that mouse mutants affecting the centrosome should exhibit heterotopias in which neurons remain ectopically attached in the region of the ventricle. Phenotypes in such mice vary depending on which centrosomal gene is targeted as well as the timing and extent of gene loss (*Buchman et al., 2010*; *Insolera et al., 2014*; *Lizarraga et al., 2010*). However, Sas4/Cenp2 mutant mouse cortex exhibits mis-localisation of mitotically stalled neural progenitors away from the ventricle and also some neuronal heterotopias (*Insolera et al., 2014*), consistent with the findings reported here following compromise of the centrosome specifically in presumptive neurons.

Cell delamination from within epithelial sheets is a fundamental cell behaviour linked to both differentiation and disease (e.g. *Kesavan et al., 2014*; *Slattum and Rosenblatt, 2014*; *Nikitas and Cossart, 2012*; *Vasioukhin, 2012*). Our data uncover novel cytoskeletal architecture and cell biological mechanisms that mediate this process in the neuroepithelium. It is important now to determine

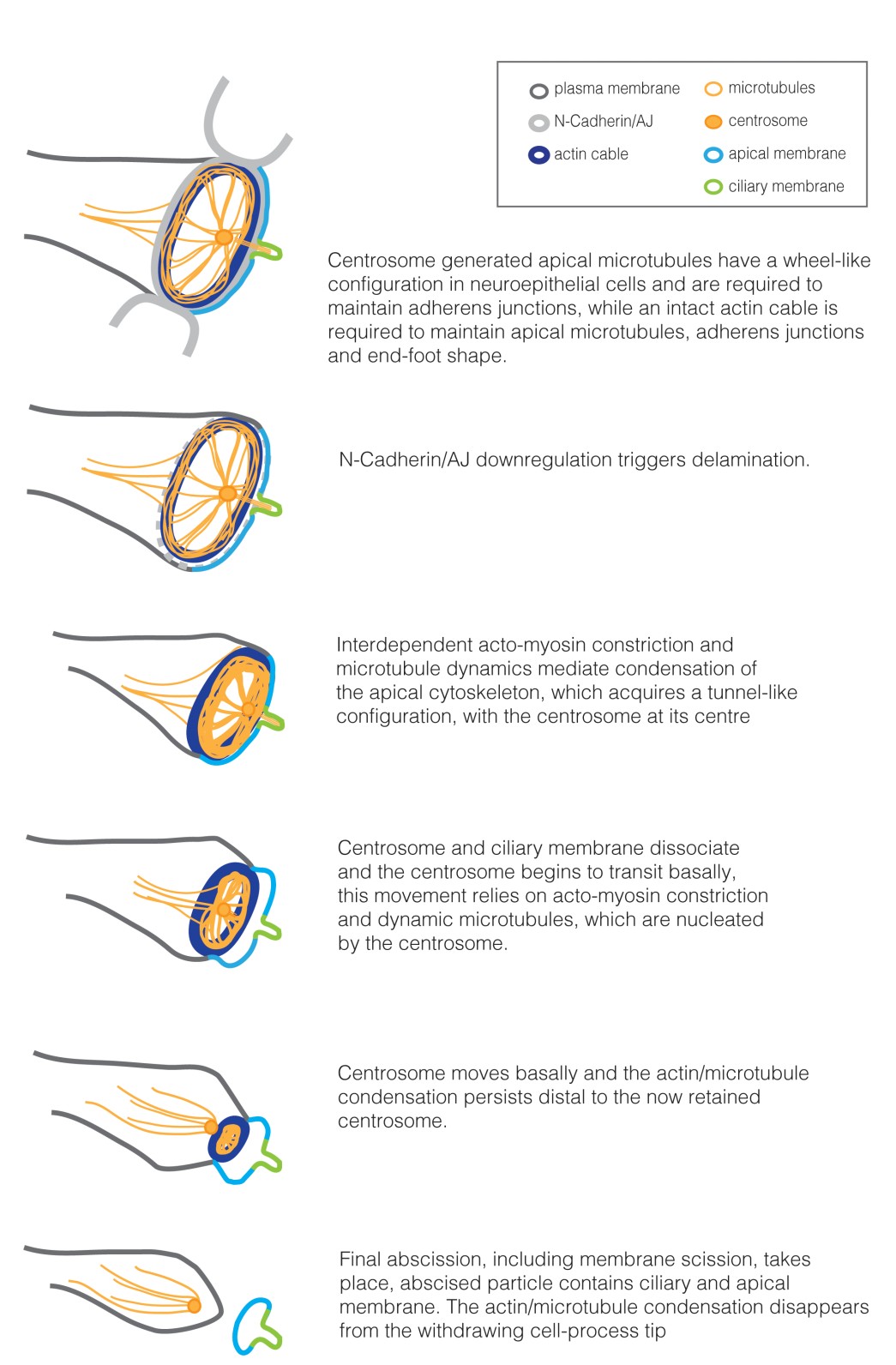

**Figure 10.** Summary of cytoskeletal configuration and dynamic changes in the apical end-foot during neuronal delamination.
DOI: https://doi.org/10.7554/eLife.26215.058

whether these cytoskeletal configurations and regulatory relationships are conserved in other cell types and if they are perturbed in pathological contexts. Indeed, our findings are consistent with recent work demonstrating microtubule network remodelling prior to centrosome reorientation in cells undergoing EMT-like polarity inversions (*Burute et al., 2017*). It therefore seems likely that the apical microtubule-actin alignment uncovered here is a common feature of epithelial cells and that the interdependency of effective acto-myosin constriction and dynamic microtubules during apical constriction is a shared mechanism which may ensure retention of the apically localised centrosome characteristic of many epithelial cell types. These findings further show that delamination is an active process downstream of AJ loss and may open up new opportunities to manipulate delamination by targeting context specific proteins that orchestrate actin and microtubule interactions.

## Materials and methods

### In ovo electroporation and plasmids

Fertilized chicken (Gallus gallus domesticus) eggs were obtained from Winter Egg Farm (Hertfordshire - Royston SG8 7RF, UK) and incubated at 38°C to Hamburger and Hamilton stages 10–12. This was followed by neural tube electroporation of plasmids as described previously (*Das et al., 2012*). Minimal plasmid concentrations were used to enable visualisation of the marker being analysed (typically within the range of 25–100 ng/µl). Only cells that expressed low levels of the markers were chosen for subsequent analysis. EMTB-GFP was a kind gift from Professor WA Harris, University of Cambridge, UK (*Norden et al., 2009*), F-tractin-mKate2 from Alwyn Dady, University of Dundee, UK, Stathmin-GFP from Lynne Cassimeris (Addgene plasmid # 86782), Drebrin-shRNA, scrambled control, Drebrin-YFP and Drebrin-mCherry constructs from Dr John Chilton (University of Exeter, UK), PACT-KillerRed was generated by replacing TagRFP in PACT-TagRFP with KillerRed on an AgeI/NotI fragment. The KillerRed construct was obtained from Evrogen (FP962).

### Immunofluorescence and fixed tissue imaging

Hamburger and Hamilton Stage 17–18 embryos were fixed in 4% paraformaldehyde and equilibrated overnight in 30% sucrose at 4°C. These were then embedded in 1.5% LB agar (Sigma, L7025) and 5% sucrose, dissolved in MilliQ water. Mounted tissue was dehydrated again for 24 hr in 30% sucrose and snap frozen on dry ice. 20 µm thick sections were then collected using a Leica cryostat (maintained at −25°C).

To visualise endogenous microtubules, the spinal cord region of E3 chick embryos were fixed with pre-warmed (37°C) PHEMO fix solution (68mMPIPES, 25mMHEPES, 15 mM EGTA, 3 mM MgCl2, 3.7% PFA, 0.05% Glutaraldehyde, 0.5% tritonX) for 40 min, washed twice with PHEMO buffer ((68 mM PIPES, 25 mM HEPES, 15 mM EGTA, 3 mMMgCl2, 10% [v/v] DMSO, pH 6, with 10M KOH) and quenched with 100 mM Glycine for 60 min (*Wagstaff et al., 2008*) before being equilibrated in 30% sucrose overnight. For immunofluorescence of EMTB-GFP, neural tubes were fixed with pre-warmed (37°C) 4% PFA for 30 min. For *en-face* imaging of endogenous microtubules, the neural tube of E3 chick or mouse embryos (E12.5) was halved sagittaly (dorsoventrally) along the ventricle and fixed in pre-chilled (−20°C) 100% methanol for 10 min at −20°C. To investigate the effect of microtubule depolymerisation and actin polymerisation inhibition on fixed tissue, neural tube explants for en face imaging were incubated in pre-warmed neurobasal medium containing nocodazole (8.5 µM, Calbiochem, CAS 31340-18-9) for 1 hr (in this explant assay, microtubule depolymerisation is not observed at 30 min Nocodazole treatment, data not shown) or latrunculin-A (1 µM, Abcam, ab144290) for 15 min (severe tissue collapse at 20 min incubation, data not shown). They were then fixed in PHEMO fix solution for 30 min and processed for immunofluorescence imaging in whole mount.

For all fixation methods, E3 embryos were handled in pre-warmed (37°C) Leibovitz's L-15 media (ThermoFisher, 11415049) to maintain microtubule integrity. E12.5 Mouse tissue for *en-face* imaging was blocked overnight with donkey anti-mouse IgG (1:200, Jackson Immunoresearch, 715-005-151). Primary antibody dilutions in blocking buffer (0.1% Triton-X-100% and 1% heat inactivated donkey serum, in PBS): Acetylated alpha tubulin (Sigma, T7451; RRID:AB_609894) 1:150, alpha tubulin (YL1/2) 1:200, alpha tubulin (Abcam, ab7291) 1:150, N-Cadherin (ThermoFisher, 13–2100; RRID:AB_2533007) 1:300, GFP (Abcam, ab6673; RRID:AB_305643) 1:500, IFT88 (Proteintech, 13967–1-AP;

RRID:AB_2121979) 1:200, γ-tubulin (Sigma, T5326;RRID:AB_532292) 1:300, Drebrin (Abcam, ab11068; RRID:AB_2230303) 1:200.

All secondary antibodies used were Alexa Fluor conjugates at 1:500 (Donkey anti-goat 488 [ThermoFisher, A-11055; RRID:AB_2534102], Donkey anti-rat 568 [Abcam, ab175475; RRID:_AB2636887], Donkey anti-rabbit 568 [ThermoFisher, A-10042; RRID:AB_2534017], Donkey anti- mouse 488 [ThermoFisher, A-21202; RRID:AB_141607]). Actin was stained with conjugated CF640R Phalloidin (Biotum, 00050). Sections were mounted on Prolong Gold antifade mountant (ThermoFisher, P36930). Neural tube expants were mounted in 0.6% low gelling temperature agarose (Sigma, A9045). Images were acquired using a 40 × 1.3 NA or 60 × 1.42 NA objective on a Deltavision Core microscope system (Applied Precision LLC, Issaquah, WA).

## Sample preparation for structured illumination and STED

Cover-slips of 0.17 mm thickness (no. 1.5) were coated with poly-l-lysine (Sigma, P8920) for 30 min at 37°C, washed twice with MilliQ water and left to dry overnight at room temperature. Cryosections of 20 μm thickness were mounted directly on the cover-slips. The EMTB-GFP and F-tractin-mKate2 fluorescent signals were amplified with anti-GFP (Abcam, ab6673) and anti-tRFP (Evrogen, AB233; RRID:AB_2571743) primary antibodies (both at 1:300), respectively. Secondary antibodies were conjugated with Alexa 488 and Alexa 568 (ThermoFisher, A-11055, ThermoFisher, A-10042). Tissue sections were mounted on Slowfade Gold antifade (ThermoFisher, S36936) or Prolong Diamond (ThermoFisher, P36965) mounting media.

## Structured illumination and STED imaging

Structured illumination microscopy was carried out on the OMX Blaze system (GE Healthcare) equipped with a UPlanSApochromat 63 × 1.42 NA, oil immersion objective lens (Olympus, Center Valley, PA), scientific CMOS camera (PCO AG, Germany) and a 488 nm solid-state laser. Samples were illuminated by a coherent scrambled laser light source that had passed through a diffraction 10.7554 10.7554 10.7554 grating to generate the structured illumination by interference of light orders in the image plane to create a 3D sinusoidal pattern, with lateral stripes approximately 0.2 μm apart. The pattern was shifted laterally through five phases and through three angular rotations of 60° for each Z-section, separated by 0.125 μm. Exposure times were typically between 10 and 50 ms, and laser power was adjusted to achieve optimal intensities of between 500 and 1000 counts in a raw image of 15-bit dynamic range, at the lowest possible laser power to minimize photo bleaching. Raw images were processed and reconstructed to reveal structures with greater resolution (*Gustafsson et al., 2008*) implemented using SoftWorx, ver. 6.0 (Applied Precision, Inc.). The channels were then aligned in x, y, and rotationally using predetermined shifts as measured using 100 nm TetraSpeck (Invitrogen) beads with the SoftWorx alignment tool (Applied Precision, Inc.).

STED imaging was carried out using a Leica Microsystems TCS SP8 STED system equipped with a 100 × 1.4 NA oil immersion STED objective. Images in the green channel were acquired using a 488 nm excitation laser and 592 nm depletion laser. Images in the red channel were acquired using a 568 nm excitation laser and 660 nm depletion laser. Z-sections were separated by 0.2 μm and images were scanned at 10 Hz using 2x line averaging. The resulting images were deconvolved using Huygens Professional (Scientific Volume Imaging).

## Embryo slice culture

Embryonic slice culture was carried out as described previously (*Das et al., 2012*). Briefly, chick neural tubes were electroporated at Hamburger and Hamilton stage 10–12 and incubated for 18 hr. Transverse spinal cord slices were obtained from the trunk region between the wing and leg buds and embedded in collagen (Corning, 354236) (supplemented with 0.1% acetic acid, 5x L-15 medium [ThermoFisher, 41300] and 7.5% sodium bicarbonate [ThermoFisher, 25080094]) in poly-D-lysine coated glass-bottomed petri-dishes (World Precision Instruments, FD35-PDL-100) as described previously. For en face imaging the same region of the neural tube was halved dorso-ventrally along the ventricle. One side was discarded and part of the other intact side (4–5 somites long) including the overlying somites was embedded, with the apical end-feet facing the glass of the dish. Slices embedded in collagen were allowed to recover for three hours in Neurobasal medium (ThermoFisher, 12348017) supplemented with B-27 (ThermoFisher, 17504044), glutamax (ThermoFisher,

35050038) and gentamicin (ThermoFisher, 15750037) at 37°C before imaging was started. For inhibitor experiments the medium was replaced with warmed medium containing one of the following small molecules at the specified final concentration or their controls: nocodazole (8.5 μM, Calbiochem, CAS 31340-18-9), taxol (10 μM, Sigma, T7191), ML-7 (20 μM, Sigma, I2764), DMSO (Sigma) or $H_2O$ at the start of imaging.

## Time-lapse imaging and processing

Time-lapse imaging of embryo slices was performed using a Deltavision Core microscope system in a WeatherStation environmental chamber maintained at 37°C. (GE Healthcare). Imaging was limited to minimal exposure times (50–100 milliseconds) to detect low fluorescence levels (*Das and Storey, 2014*; *Das et al., 2012*). Image acquisition was performed using an Olympus 40 × 1.3 NA oil immersion objective or an Olympus 40 × 1.25 NA silicone oil immersion objective, a solid stated LED light source and a CoolSnap HQ2 cooled CCD camera (Photometrics). Unless otherwise stated, 33–34 optical sections spaced 1.5 μm apart were acquired for each slice at 5–10 min intervals (exposure time 5–50 milliseconds for each channel, 512 × 512 pixels, 2 × 2 binning). For en face imaging of EB3-GFP comets, 5–8 optical sections spaced 0.5 μm apart were acquired at ~1.5–3.0 s intervals (exposure time of 150–200 milliseconds for the EB3-GFP comets). For the KillerRed and its control experiments, each slice was exposed to a total of 15 min of green light irradiation. Images were deconvolved using the SoftWorx image processing software. The position of the apical surface at each time point was monitored by acquiring a bright-field reference image at the middle of the z-stack.

## Measurement between EB3-GFP and F-tractin-mKate2 inter-peak distance

Trail movies of EB3-GFP comets were generated out using the SoftWorx image processing software. For the measurement of EB3-GFP and F-tractin-mKate2 inter-peak distance, a line of 1 μm was drawn across the EB3-GFP comet and fluorescent intensities measurements were carried for both GFP and mKate2 using the FIJI version of the ImageJ software suite (*Schindelin et al., 2012*). The data were then fitted to Guassian curves on FIJI (Analyse→ Tools→ Curve Fitting) and the distance between each EB3-GFP and F-tractin-mKate2 pair calculated where fluorescent intensity was the highest (inter-peak distance, *Figure 2D–F*).

## Measurement of fluorescence intensities and area under the curve

All measurements of fluorescent intensities were carried out using the FIJI software (*Schindelin et al., 2012*). For proper comparison of fluorescence intensities in *Figure 3*, the same exposure times were used for DMSO control and the small molecule treatments. For the measurement of the grey scale values of N-Cadherin or actin fluorescence intensity in *Figure 3*, a straight line of 2 μm (Latrunclin-A experiments) or 4 μm (Nocodazole experiments) μm was drawn across the adhesion belt of two cells. Background fluorescence, using the freehand tool, was obtained by measuring the mean grey scale value of the area of one of the cells, defined by the N-Cadherin localisation (excluding the adhesive belt region). The same N-Cadherin defined area was used to obtain the measurement of the mean grey scale value of tubulin fluorescence. Furthermore, the N-Cadherin localisation was used, including the adhesive belt region, to measure the end-foot area (polygon tool). Mean background fluorescence for tubulin was obtained by taking measurements within mitotic cells, before reaching the mitotic microtubules along the Z-axis.

For *Figure 1—figure supplement 1* and *Figure 7C*, a straight line (of 2 and 1 μm, respectively), across the region of interest was used for the measurement of the grey scale values. The values for each channel were then normalised to the highest value set as 1. Graphs were plotted accordingly.

To calculate the area under the curve in *Figure 3C, D, H and I* the following formula was used, (Y1 + Y2)/2 *dx where Y1 is the normalised fluorescence intensity at one point, Y2 is the normalised fluorescence intensity of the following point and dx is the distance, defined by the pixel size. For each cell, the total area under the curve is calculated by adding all the values obtained. For the area that corresponds to the adhesion belt, the middle ten values were added.

For *Figure 6*, presumptive neurons in the right configuration for abscission, mis-expressing pCIG-Neurog2, were used for the fluorescence intensity measurements. As established, the majority of

such cells, treated with Taxol or ML-7 do not abscise and the fluorescence intensity levels of EMTB-GFP or F-tractin-mKate2 were compared to cells in DMSO conditions. The mean grey value of fluorescence intensity, on maximum intensity projections, was measured every thirty minutes and normalised to background levels. For control treatments, the seventh measurement corresponded to the abscission time (0 min).

### Statistical analysis

The mean inter-peak distance (*Figure 2F*) was compared between time-points using the paired t-test. The t-test was used to compare the mean area under the curve, the normalised tubulin fluorescence intensity and the apical end-foot area between treatments for *Figure 3*. The values obtained for each of the above measurements are expected to follow a normal distribution (continuous data). In *Figure 6*, comparisons of normalised fluorescent intensity trends between small molecule treatments and their respective controls, over time, were performed on SigmaPlot software using the 2-way ANOVA test.

## Acknowledgements

We thank Dr Paul Appleton and the Dundee Imaging Facility for advice and technical assistance with OMX Super-resolution imaging and Dr Marek Gierlinski (Data Analysis Group) and Dr Graeme Ball (Dundee University imaging facility) for advice on data analysis and statistics. We are also grateful to Dr Jens Januschke and Professors Kees Weijer and Inke Nathke for insightful comments on this manuscript. This work was supported by a senior Investigator award from the Wellcome Trust to KGS (WT102817AIA). The study also utilized microscopy resources supported by a Wellcome Trust multi-user equipment grant for the development of tissue imaging approaches (WT101468) and a Medical Research Council Next Generation Optical Microscopy Award (MR/K015869/1). STED imaging was carried out at the University of Manchester Bioimaging facility.

## Additional information

### Funding

| Funder | Grant reference number | Author |
|---|---|---|
| Medical Research Council | Medical Research Council Next Generation Optical Microscopy Award (MR/K015869/1) | Kate G Storey |
| Wellcome Trust | Multi-User Equipment Award (WT101468) | Kate G Storey |
| Wellcome Trust | WT102817AIA | Kate G Storey |

The funders had no role in study design, data collection and interpretation, or the decision to submit the work for publication.

### Author contributions

Ioannis Kasioulis, Investigation, Visualization, Methodology, Writing—original draft, Writing—review and editing, Carried out most of the experiments; Raman M Das, Conceptualization, Data curation, Supervision, Investigation, Visualization, Methodology, Writing—original draft, Writing—review and editing, Carried out some of the experiments; Kate G Storey, Conceptualization, Resources, Data curation, Supervision, Funding acquisition, Visualization, Methodology, Writing—original draft, Project administration, Writing—review and editing

### Author ORCIDs

Ioannis Kasioulis https://orcid.org/0000-0001-5054-5357
Raman M Das http://orcid.org/0000-0003-3375-619X
Kate G Storey http://orcid.org/0000-0003-3506-1287

**Decision letter and Author response**
Decision letter https://doi.org/10.7554/eLife.26215.060
Author response https://doi.org/10.7554/eLife.26215.061

## Additional files

**Supplementary files**
• Transparent reporting form
DOI: https://doi.org/10.7554/eLife.26215.059

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
