## [Decision Letter]

Thank you for submitting your article "Apical microtubules interact with the actin cable to orchestrate centrosome retention and neuronal delamination" for consideration by *eLife*. Your article has been reviewed by three peer reviewers, and the evaluation has been overseen by a Reviewing Editor and Anna Akhmanova as the Senior Editor. The following individual involved in review of your submission has agreed to reveal their identity: Phillip Gordon-Weeks (Reviewer #1).

The reviewers have discussed the reviews with one another and the Reviewing Editor has drafted this decision to help you prepare a revised submission.

Summary:

The manuscript by Kasioulis, Das and Storey examines the role of microtubule and actin interaction in neuronal delamination from a chick spinal cord germinal zone niche. The authors use time-lapse imaging to document cytoskeletal rearrangement during delamination and hypothesize that microtubule/actin cooperation underlies a series of events occurring during delamination. Ultimately, the authors propose that acto-myosin contraction coupled with microtubule dynamics creates a centrosome/microtubule translocation event essential for delamination.

The experiments are well designed and the results for the most part convincing. Conceptually, the idea that microtubule-actin interactions control a complex morphogenic event is interesting and rather timely. In general, the field of neuronal morphogenesis focuses on either the activity of microtubules and actin in isolation. Thus, the implications of the study are very important to an area that has struggled to analyze higher order cytoskeletal integration.

Essential revisions:

1) Key genetic manipulations are needed to bolster some of the experiments done only with cytoskeletal drugs. While drugs like nocodazole, taxol or ML7 are accepted in the field they possess limitations: they can have off-target consequences and in this situation treating an entire tissue with drugs will have non-cell autonomous impacts on the experiments in question. This requires additional manipulation of the microtubule and actin cytoskeleton. Over-expression of Stathmin, which depolymerizes microtubules, and a Dominant Negative Rock Kinase construct, which inhibits actomyosin contractility, are methods that have used by others to cell autonomously perturb the relevant cytoskeletal systems. Addition to Figure 6 and Figure 7 are sufficient for such experiments.

2) Currently, some claims regarding cytoskeletal interactions are on balance descriptive. Thus, conclusions and the Figure 9 model should be moderated in the cases where the cytoskeletal relationship are only defined by correlative imaging. Most imaging of the study is done with confocal microscopy, which has a poor axial resolution to detect such interactions. Therefore, claims of cytoskeletal interaction should also be buttressed by a mechanistic manipulation of a candidate microtubule-actin linking factor. A cell permeable IQGAP1 inhibitor (Oblander SA, Brady-Kalnay SM. Mol Cell Neurosci. 2010 May;44(1):78-93) or a drebrin shRNA (Dun et al. Mol Cell Neurosci. 2012 Mar;49(3):341-50) are useful tools validated in chick for these studies. Addition to Figure 6 and Figure 7 are sufficient for such experiments.

3) Additional statistical analysis would add strength to many of the author's claims beyond some of the single examples shown. In most cases, the degree of delamination is only stated as a percentage of imaged cells, a measure that does not reveal mechanistic subtly of possible phenotypes. Measuring the soma directed movement of apically anchored processes in existing data as a time course is an appropriate surrogate for delamination of processes in Figure 5, Figure 7 and 8. Statistical representation of the relationships in existing data displayed in Figure 1, Figure 2 is needed. Finally, key experiments such as those presented in Figure 6 have very low experimental observations compared to other figures where 30-50 cells were examined.

---

## [Author Response]

Essential revisions:1) Key genetic manipulations are needed to bolster some of the experiments done only with cytoskeletal drugs. While drugs like nocodazole, taxol or ML7 are accepted in the field they possess limitations: they can have off-target consequences and in this situation treating an entire tissue with drugs will have non-cell autonomous impacts on the experiments in question. This requires additional manipulation of the microtubule and actin cytoskeleton. Over-expression of Stathmin, which depolymerizes microtubules, and a Dominant Negative Rock Kinase construct, which inhibits actomyosin contractility, are methods that have used by others to cell autonomously perturb the relevant cytoskeletal systems. Addition to Figure 6 and Figure 7 are sufficient for such experiments.

We have made substantial efforts to address this request for use of complementary genetic approaches to test the role of microtubules and acto-myosin contractility on a cell by cell basis. To achieve this, plasmids are injected into the lumen of the neural tube and electroporation across the embryo leads to transfection of cells – most often these are mitotic cells which present the largest surface area at the lumen/apical surface. These cells will have first to make the proteins encoded by these constructs and in many cases will have re-entered the cell cycle and require transit through mitosis before differentiation. This reflects the nature of this tissue, it is a proliferative neuroepithelium in which neurons are born and it is this process that we are studying. As we are sure you will appreciate, we have found that introducing constructs that interfere with microtubules or acto-myosin and monitoring their effects in cells poised to delaminate has been difficult; we have succeeded for microtubules, but not for acto-myosin.

a) We have mis-expressed Stathmin-GFP and found that this reduced delamination of cells poised to abscise; this new data is presented in the last paragraph of the subsection “Microtubules are required for neuronal delamination” and in Figure 5—figure supplement 3) (Figure 5—video 7 and Figure 5—video 8). Targeting Stathmin-GFP to cells poised to delaminate required significant optimization and was achieved eventually by carrying out many transfections at low plasmid concentrations to be able to monitor sufficient cells in this state for this analysis. The data we present indicates that this genetic approach replicates the results generated using taxol or nocodazole and validates use of these small molecules to investigate the role of microtubules in this assay. Given the difficulties we encountered targeting this construct to cells about to differentiate we did not extended this approach to assessing effects of Stathmin-GFP on EMTB, F-tractin and centrosome dynamics.

b) Experiments to genetically manipulate acto-myosin contractility in presumptive neurons have proved particularly challenging. Initially, we transfected cells with mKate2-GPI, Ngn2-IRES-mApple and a dominant negative Rock Kinase-GFP fusion construct to inhibit acto-myosin contractility. This construct was driven by a promoter that works weakly in avian cells (CMV). Upon imaging we were unable to detect significant levels of GFP expression, although transfected progenitor cells expressing mKate2-GPI, Neurog2-IRES-mApple were seen undergoing mitosis and later neuronal delamination. The latter observation suggested that expression of this plasmid, at such low levels, was unable to interfere with acto-myosin constriction in this assay. We therefore next used a further construct with a more active promoter. This involved use of a non-activatable version of myosin light chain 2 (MRLC2^T18AS19A^-GFP) (which attenuates actin constriction and mimics effects of blebbistatin, Miklavc et al. 2015, JCS 128, 1193-1203). As expression from the MRLC2^T18AS19A^-GFP construct is driven by the CAGGS promoter, which induces robust expression in avian cells, we expected to observe higher levels of MRLC2^T18AS19A^-GFP. However, when transfected tissue was imaged, we found very few transfected cells amidst a background of cell debris, suggesting that mis-expression of this construct has a detrimental effect on cell viability. Indeed, we managed to image a small number of cells (6 cells, 6 slices, 6 embryos) which were able to initiate mitosis, but quickly experienced cell death following this. We conclude that to disrupt acto-myosin contractility in cells poised to delaminate it is necessary to use methods that allow very precise temporal control. At present, the most efficient approach available to achieve this is the use of chemical inhibitors, whose application to the imaging medium can be controlled in a precise manner.

In conclusion, we have addressed the requested experiments and we have been able to show that genetic interference with microtubule polymerisation inhibits neuronal delamination. We have been unable to extend this genetic approach to manipulation of acto-myosin activity, in large part because we could not interfere with this process without also targeting the preceding mitosis. It is informative here that Stathmin-GFP does not affect progression through mitosis due to its increased phosphorylation during this phase of the cell cycle (Brattsand et al. 1994, Eur J Biochem 220, 359-368; Luo et al. 1994, JBC 14, 10312-10318; Gavet, O et al. 1998, JCS 111, 3333-3346). Recent experiments where Stathmin was over-expressed in the zebrafish retina showed similar effects on newborn neurons while sparing mitosis (Icha et al. 2016, JCB 215, 259-275). This may explain why we were eventually able to target prospective neurons with low level Stathmin-GFP, but could not use this mis-expression approach with an acto-myosin interfering construct. So, while we have been unable to carry out all the experiments requested here, this is in large part due to the biological constraints inherent in this tissue based assay.

2) Currently, some claims regarding cytoskeletal interactions are on balance descriptive. Thus, conclusions and the Figure 9 model should be moderated in the cases where the cytoskeletal relationship are only defined by correlative imaging.

We have moderated conclusions relating to actin and microtubule interaction and the text in Figure 10 (formerly labeled as Figure 9) as requested. We have shown that microtubules are required for acto-myosin constriction and that conversely that acto-myosin activity is required for normal microtubule dynamics; these two cytoskeletal components are therefore inter-dependent during delamination, but we agree that we have not identified direct interactions. We have accordingly replaced “interact” with phrases indicating an inter-dependent regulatory relationship. This includes changing the title to “Inter-dependent apical microtubule and actin dynamics orchestrate centrosome retention and neuronal delamination”, the last sentence of the Abstract and of the Introduction, first and last lines of the Discussion and the final paragraph of the Discussion.

Most imaging of the study is done with confocal microscopy, which has a poor axial resolution to detect such interactions.

There is a mis-understanding here. We do not use confocal microscopy, but rather a widefield Δ-Vision imaging system. We take images at 1.5uM intervals through a 50uM deep Z stack and these are subject to deconvolution to remove out of focus fluorescence. These data are presented as maximum intensity projections in images provided for figures, but many movies are also provided. Although the axial resolution achieved through widefield microscopy is modest, this technique facilitates image acquisition at superior signal-to-noise, enabling us to capture very fine and dim sub-cellular structures in living tissue. Capturing such detail is often impossible using the other available techniques such as confocal or other super-resolution techniques due to photo-bleaching of these dim and dynamic signals. We do complement this approach with fixed super-resolution techniques (SIM and STED) to confirm the wheel-like microtubule configuration and the close association between sub-apical actin and microtubules during neuronal delamination. Combined, these approaches provide a detailed view of the cytoskeletal re-arrangements that occur during the process we are studying.

Therefore, claims of cytoskeletal interaction should also be buttressed by a mechanistic manipulation of a candidate microtubule-actin linking factor. A cell permeable IQGAP1 inhibitor (Oblander SA, Brady-Kalnay SM. Mol Cell Neurosci. 2010 May;44(1):78-93) or a drebrin shRNA (Dun et al. Mol Cell Neurosci. 2012 Mar;49(3):341-50) are useful tools validated in chick for these studies. Addition to Figure 6 and Figure 7 are sufficient for such experiments.

We were unable to source the reported cell permeable IQGAP1 inhibitor. We have, however, carried out extensive experiments to characterize Drebrin localization and its relationship to actin and microtubules in the developing neuroepithelium. We found that Drebrin has a widespread cytoplasmic distribution in neuroepithelial cells and that it co-localises with the sub-apical actin belt in the apical end-foot, subsection “The actin and microtubule cross-linking protein Drebrin is required for neuronal delamination”, Figure 7. Knockdown of Drebrin using short hairpin RNA resulted in a reduction in the number of cells delaminating; Drebrin shRNA (4/27- 15%) and Scrambled sh-control (7/10- 70%), Figure 7—figure supplement 1. We were not able to extend this analysis to monitor EMTB and centrosome dynamics in the presence of Drebrin shRNA because it is difficult to live-image three different fluorescent labels in the same experiment (EMTB or centrosome or actin marker, membrane marker, Neurog2-NLS-GFP and Sh-Drebrin-GFP) and given this technical complexity we felt these further experiments were beyond of the scope of this paper. The experiments we do present here identify Drebrin as a potential microtubule-actin linking factor expressed at the right time and place to regulate neuronal delamination and demonstrate its requirement for this process.

3) Additional statistical analysis would add strength to many of the author's claims beyond some of the single examples shown. In most cases, the degree of delamination is only stated as a percentage of imaged cells, a measure that does not reveal mechanistic subtly of possible phenotypes. Measuring the soma directed movement of apically anchored processes in existing data as a time course is an appropriate surrogate for delamination of processes in Figure 5, Figure 7 and Figure 8.

In a sense, we already use nuclear position to define cells that are about to abscise. We mis-express Neurog2 (raised levels of which promote neuronal differentiation) and we then look for and monitor cells with a basally located nucleus that possess an attached apical cell-process. As we monitor these cells we can see whether the nucleus changes position (i.e. moves apically and so has re-entered the cell cycle – this is not seen in *Neurog2* mis-expressing cells in this configuration). Inclusion of cells that have a basally moving nucleus would increase our numbers, but there is a possibility that some of these cells might re-enter the cell cycle beyond the imaging period. We would prefer to only include cells with a static basally positioned nucleus. As described below, we have carried out further experiments to increase the number of cells for some experiments.

Statistical representation of the relationships in existing data displayed in Figure 1, Figure 2 is needed.

We have carried out the detailed statistical analyses requested by the reviewers. This new data is now presented in the second paragraph of the subsection “A wheel-like microtubule configuration in the neuroepithelial cell apical end-foot” and in the subsection “The centrosome nucleates microtubules which radiate towards and extend along the actin cable”. We have measured the alignment of actin and tubulin at the level of the adherens junctions in fixed cell data, determining the number of cells in which this near coincident; we have used the “trail movies” feature of the SoftWorx software to track EB3-GFP movements over time and so provide further evidence for microtubule growth trajectories; and we have used these trail movies to measure inter-peak distance of EB3-GFP and F-tractin-mKate2 to provide a statistical analysis of the relationship between growing microtubules and the actin cable.

Finally, key experiments such as those presented in Figure 6 have very low experimental observations compared to other figures where 30-50 cells were examined.

We have carried out further experiments and increased the number of cells monitored. This new data is now presented in the second and last paragraphs of the subsection “Apical microtubule and actin conformational dynamics are inter-dependent in delaminating cells”.